# Maternal asthma imprints fetal lung ILC2s via glucocorticoid signaling leading to worsened allergic airway inflammation in murine adult offspring

**Tomoaki Takao**[1,2], **Ako Matsui**[1], **Chie Kikutake**[3], **Keiko Kan-o**[4], **Azusa Inoue**[5,6], **Mikita Suyama**[3], **Isamu Okamoto**[2] & **Minako Ito**[1] ✉

The root of asthma can be linked to early life, with prenatal environments influencing risk. We investigate the effects of maternal asthma on the offspring's lungs during fetal and adult life. Adult offspring of asthmatic mothers show an increase in lung group 2 innate lymphoid cell (ILC2) number and function with allergen-induced lung inflammation. Offspring of asthmatic mothers show phenotypic alteration of their lung ILC2s during fetal life, with increased expression of genes related to activation and glucocorticoid signaling. Furthermore, these offspring carry overlapping chromatin-accessible altered regions, including glucocorticoid receptor-binding regions in their lung ILC2s both at the fetal stage and adulthood, suggesting persistent prenatal epigenetic changes. Moreover, maternal exposure to glucocorticoids has similar effects on fetal lung ILC2s and contributes to allergen-induced lung inflammation during adulthood. Thus, asthma during pregnancy may have long-term effects on lung ILC2s in the offspring from the embryonic period, contributing to an increased risk of developing asthma.

Asthma is a major health burden, affecting over 260 million people worldwide[1]. The number of patients is increasing worldwide[2], and there remains an unmet need regarding the elucidation of the mechanisms of pathogenesis and prevention of the disease[3]. The risk of developing asthma is present early in life. Epidemiological data have linked prenatal environmental factors, including smoking, stress, and maternal asthma, to a higher risk of asthma in children[4–12]. Among these factors, asthma is a common condition affecting ~17% of pregnant women[13], with one-third of them experiencing a worsening of symptoms[14]. Clinical and experimental evidence suggests that asthma during pregnancy increases the incidence of asthma in the offspring[11,12,15,16].

The effects of asthma during pregnancy on the offspring are partly responsible for the *in-utero* effects on the fetus. This is supported by the fact that maternal asthma is associated with a greater risk of developing asthma in the offspring than paternal asthma[17], and that asthma control during pregnancy reduces the risk of asthma in the offspring[18]. Previous studies have suggested that this could be attributed to potential mechanisms, including maternal programming regulating the offspring's lung immune function[19,20], and the involvement of epigenetic mechanisms[21]. However, how asthma during pregnancy specifically affects the fetus and how the effects are related to the development of asthma in the offspring are not fully understood.

[1]Division of Allergy and Immunology, Medical Institute of Bioregulation, Kyushu University, Fukuoka, Japan. [2]Department of Respiratory Medicine, Graduate School of Medical Sciences, Kyushu University, Fukuoka, Japan. [3]Division of Bioinformatics, Medical Institute of Bioregulation, Kyushu University, Fukuoka, Japan. [4]Department of Respiratory Medicine, Tokyo Women's Medical University, Tokyo, Japan. [5]Laboratory for Epigenome Inheritance, RIKEN Center for Integrative Medical Sciences, Yokohama, Japan. [6]Tokyo Metropolitan University, Hachioji, Japan. ✉e-mail: minakoito@bioreg.kyushu-u.ac.jp

Previous studies have suggested that the child does not need to acquire the same antigen-specific mechanism as the mother in the intergenerational transmission of asthma[15,22] suggesting the involvement of innate immune mechanisms. While the critical contribution of T helper 2 (Th2) cells, which represent adaptive immunity, in allergic inflammation, including asthma, is well established[23], the important role of Group 2 innate lymphoid cells (ILC2s), which lack antigen receptors, has also been increasingly emphasized in recent years[24]. ILC2s respond antigen-nonspecifically to epithelial cell-derived cytokines, including interleukin (IL)-33, IL-25, and thymic stromal lymphopoietin (TSLP), and produce key cytokines such as IL-5 and IL-13, which play crucial roles in type 2 inflammation[25–29]. In addition to these aspects of the early response to allergic inflammation, ILC2s also exhibit memory-like properties in response to stimuli[30,31]. This property, which can be described as a kind of "trained immunity," involves epigenetic changes[31]. ILC2s have also been reported to enhance Th2 cell-mediated adaptive type 2 inflammation[32], and may also play an important role in the persistence of chronic allergic inflammation.

Parabiosis experiments have shown tissue ILC2s to be tissue-resident immune cells with minimal replacement by circulating cells at a steady state[33–35]. In addition, fate mapping experiments have shown that lung ILC2s represent a heterogeneous population with three origins – embryonic, neonatal, and adult. In mouse models of *Nippostrongylus brasiliensis* infection, it is not the ILC2s supplied during infection that expand in the lungs but rather the pre-existing ILC2 pools[35]. This suggests that prenatal lung ILC2s may, at least in part, influence the phenotype of pulmonary ILC2s in postnatal individuals.

In this study, we investigate the effects of maternal asthma on the offspring's lungs during fetal and adult life. Our results show that the impact of maternal asthma on fetal lung ILC2s during pregnancy is maintained for a long term, leading to increased lung ILC2 reactivity and allergen-induced lung inflammation in adulthood. Adult offspring of asthmatic mothers show increased pulmonary ILC2 function and an altered phenotype from the embryonic stage. ILC2s in the fetal and adult lungs of offspring of asthmatic mothers show partial overlap of accessible chromatin regions altered by maternal asthma, suggesting the persistence of prenatal epigenetic changes. Our findings suggest that asthma during pregnancy has a long-term effect on lung ILC2s in the offspring from the embryonic period, contributing to an increased risk of developing asthma.

## Results

### Adult offspring of asthmatic mothers exhibit increased allergen-induced lung inflammation independent of maternal antigens

Several mouse models have been proposed in which asthma during pregnancy worsens allergen-induced lung inflammation in the offspring. However, there are conflicting reports as to whether antigen identity between mother and offspring is necessary[15,16,19,20,22]. It has also been reported that the same antigen in the mother and child can reduce asthma[36,37]. To assess the antigen-non-specific effects of asthma during pregnancy on the offspring, we induced asthma in the mothers with ovalbumin (OVA) and then modeled mild asthma in the adult offspring with mite antigen and compared them with adult offspring of non-asthmatic mothers (Fig. 1a). Adult offspring of OVA-asthmatic mothers had increased lung eosinophil and ILC2 counts in response to mite antigens compared to adult offspring of control mothers (Fig. 1b). To observe whether these changes reflected a functional disease state, we performed airway hyperresponsiveness testing. The results revealed increased airway reactivity to acetylcholine in adult offspring of asthmatic mothers (Fig. 1c, d, Supplementary Fig. 1a and Supplementary Data 1). We further confirmed the increased tissue inflammation by hematoxylin-eosin (H&E) staining of lung sections (Fig. 1e). Upon investigating epithelial-derived cytokines, we found that IL-33 levels were higher in the lungs of adult offspring from asthmatic mothers during mite antigen exposure (Supplementary Fig. 1b). These

suggest that there is a mother-to-child antigen non-specific pathway of allergen-induced lung inflammation in adult offspring of mothers who developed asthma during pregnancy.

### Lung ILC2s from adult offspring of asthmatic mothers shows increased reactivity

Adult offspring of asthmatic mothers were associated with a significant and selective increase in the numbers of eosinophils and ILC2s in the lungs compared to those of control mothers (Fig. 1b), as well as an increased number of activated ILC2s that secreted cytokines such as IL-5 and IL-13 (Fig. 2a). To confirm whether similar changes in ILC2s are also observed in asthma models with Th2-driven immunity, we conducted experiments in which adult offspring were sensitized and exposed to mite antigen (Supplementary Fig. 2a). The data revealed a trend toward increased eosinophil numbers in the lungs of adult offspring from asthmatic mothers. Additionally, the number of ILC2s in the lungs was significantly elevated, along with an increase in cytokine-producing ILC2s. In contrast, while Th2 cells and cytokine-producing Th2 cells in the lungs showed an upward trend, these changes were not statistically significant. These results suggest that even with the sensitization and exposure protocol to mite antigen, ILC2s may be robustly activated in the lungs of adult offspring from asthmatic mothers. To assess whether this increase in the number and activation of lung ILC2s was a cell-autonomous function of the ILC2s themselves or whether it resulted from a change in the response of the lung tissue environment during mite antigen exposure, we sorted adult lung ILC2s from adult offspring of asthmatic or control mothers and cultured them in the presence of IL-33 to assess their reactivity (Fig. 2b). When ILC2s sorted from lungs of adult offspring of asthmatic mothers or controls not exposed to mite antigens were compared in culture, the levels of IL-5 and IL-13 were more elevated in the culture supernatants of lung ILC2s from adult offspring of asthmatic mothers (Fig. 2b). Flow cytometry showed no significant changes in the numbers of lung ILC2s, the proportion of IL-5+ IL-13+ ILC2s, or the median fluorescence intensity (MFI) values for IL-5 and IL-13 in adult offspring from asthmatic mothers, although a slight increase was observed (Supplementary Fig. 2b–d). To further confirm the altered responsiveness of lung ILC2s, lung ILC2s from adult offspring of asthmatic or control mothers was adoptively transferred into ILC2-deficient *Il7r*[(-/-)] mice that had previously received CD4 and CD8A neutralizing antibodies. These recipient mice were subsequently treated intranasally with IL-33 (Fig. 2c). The number of lung ILC2s in recipients did not change in both groups (Supplementary Fig. 2e). In contrast, the numbers of IL-5+ IL-13+ ILC2s and eosinophils increased in recipient mice with adult offspring of asthmatic mother-derived lung ILC2s, indicating an enhanced response to IL-33 stimulation (Fig. 2c). These results support the idea that lung ILC2s are overactive in adult offspring of asthmatic mothers.

### Transient removal of fetal lung ILC2s in asthmatic mothers alleviates eosinophilic pneumonia in adult offspring

Immune cells in the fetal lung are predominantly embryonic macrophages[38,39], but ILC2s are already present, and some fetal lung ILC2s persist; this ILC2 number is maintained in the adult lung[35]. We investigated the possibility that changes in the embryonic lung environment may contribute to worsened allergen-induced lung inflammation in adult offspring. Analysis of the fetal lungs at 18 days of gestation showed increased eosinophil and ILC2 counts in the fetal lungs of OVA-asthmatic mothers, as well as a trend toward increased IL-5+ ILC2 and IL-13+ ILC2 counts (Fig. 3a). This suggests that asthma during pregnancy leads to changes in the immunological environment, including alterations in ILC2s, in the fetal lung.

We then investigated whether removing fetal lung ILC2s, once affected by asthma during pregnancy, could ameliorate the excessive lung ILC2 response in adulthood. Maternal administration of anti-IL-

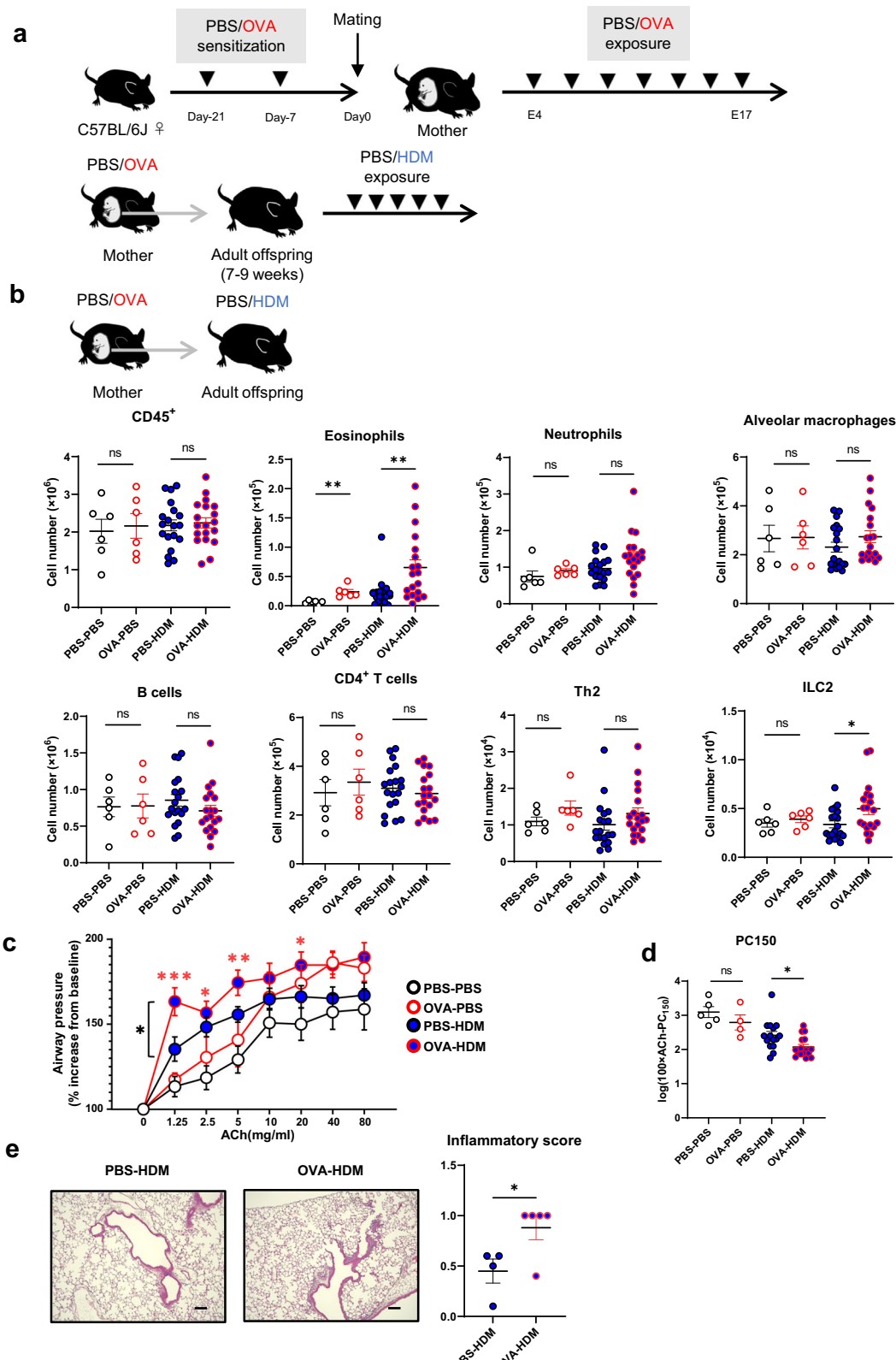

7Rα antibodies during late pregnancy results in a decrease in lung ILC2 numbers in the neonatal period but recovery by 9 weeks of age[35]. Therefore, isotype or anti-IL-7Rα antibodies were administered late in pregnancy to mothers with asthma to transiently remove fetal lung ILC2s (Fig. 3b). We confirmed the asthma in the OVA/Alum model mice was not alleviated by the anti-IL-7Rα treatment from another experiment using non-pregnant female mice (Supplementary Fig. 3a). In

addition, we observed that the administration of anti-IL-7Rα to asthmatic mothers leads to a significant decrease in ILC2 numbers in their offspring at the age of 1 week, as previously reported (Supplementary Fig. 3b). When the response to mite antigen was assessed in adult offspring of asthmatic mothers treated with anti-IL-7Rα antibodies, eosinophil and ILC2 counts and cytokine-producing ILC2 counts were reduced compared with that in adult offspring of asthmatic mothers

**Fig. 1 | Maternal asthma worsens offspring lung inflammation in an antigen-non-specific manner. a** Model of chronic maternal asthma during pregnancy: Female mice were sensitized twice with vehicle (PBS) or ovalbumin (OVA), then mated and treated intranasally with PBS or OVA intermittently from day 4 to day 17 of gestation. The adult offspring were intranasally treated with PBS or house dust mite (HDM) for 5 consecutive days at 7–9 weeks of age and subsequently analyzed. **b** Changes in the number of immune cells in the lungs of adult offspring of asthmatic or control mothers. **c** Assessment of acetylcholine-induced airway hyperresponsiveness in adult offspring of asthmatic or control mothers. **d** The log of the dose of acetylcholine required for a 150% increase in airway pressure over the baseline (log PC150 in mg/mL) was calculated for each mouse. **e** H&E staining of

lung tissues (bar, 100 μm) and Inflammation scores. In (**b**–**e**), data were pooled from three independent experiments (**b**), each experiment with one or two pregnant dams, or four independent experiments (**c**, **d**), each experiment with two pregnant dams. In (**e**), data were from one experiment, with two or three pregnant dams per group. Sample sizes were as follows: **b** PBS-PBS: $n = 6$, OVA-PBS: $n = 6$, PBS-HDM: $n = 19$, OVA-HDM: $n = 19$; **c**, **d** PBS-PBS: $n = 5$, OVA-PBS: $n = 4$, PBS-HDM: $n = 15$, OVA-HDM: $n = 16$; **e** PBS-HDM: $n = 4$, OVA-HDM: $n = 5$. In (**b**, **d**, **e**), each dot represents an individual mouse. In (**b**–**e**), data are presented as the mean ± SEM. *$p < 0.05$; **$p < 0.01$; ***$p < 0.001$; ns, not significant [unpaired two-tailed Student's $t$-test in (**b**, **d**, **e**) and two-way ANOVA followed by Tukey's test in (**c**)].

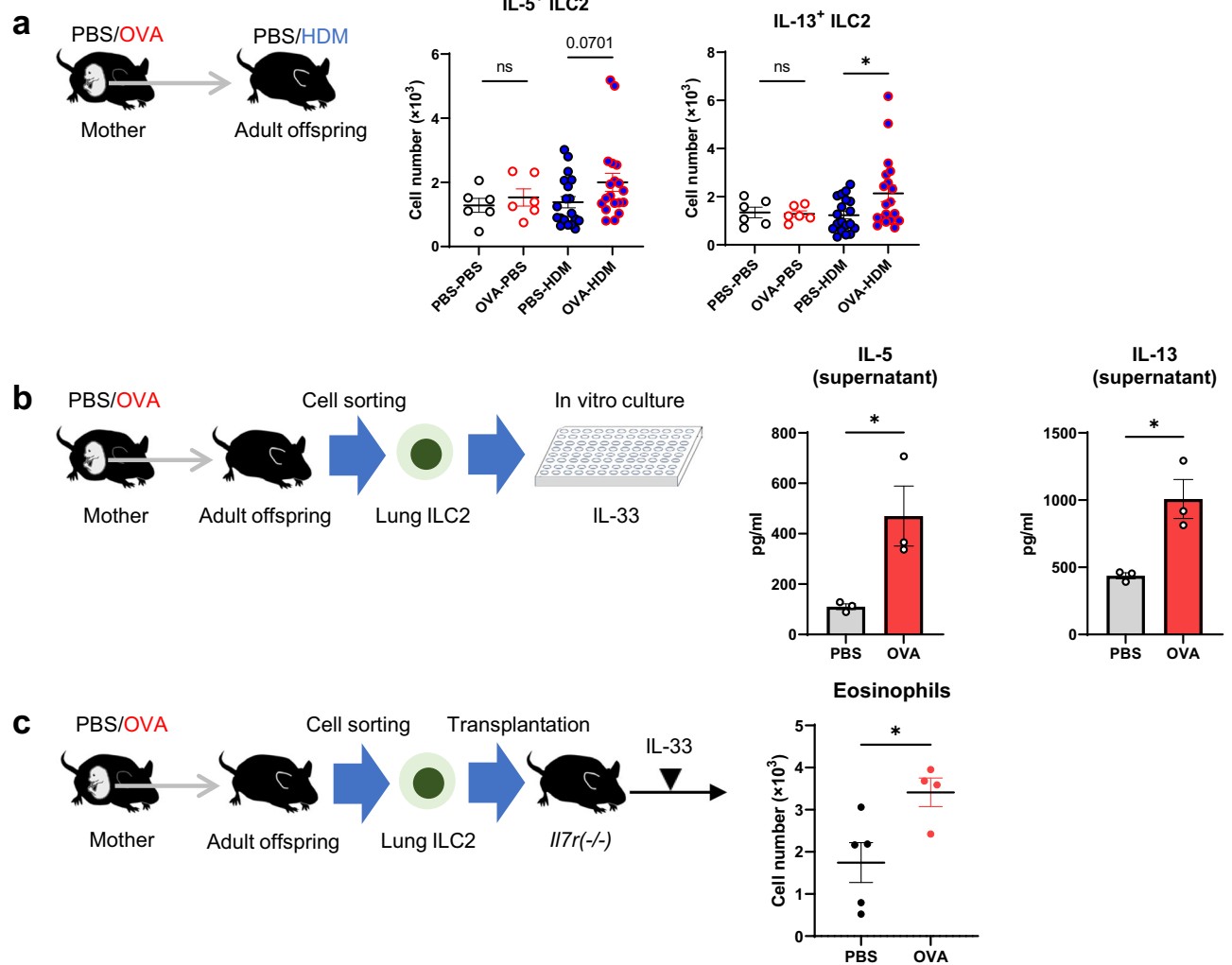

**Fig. 2 | Maternal asthma enhances lung ILC2s in adult offspring. a** Changes in the number of cytokine-producing ILC2s in the lungs of adult offspring of asthmatic or control mothers. ILC2s were stimulated with PMA/ionomycin for 4 h, and cytokine production was evaluated using intracellular staining. **b** Lung ILC2s were sorted from adult offspring of asthmatic or control mothers and cultured in the presence of IL-33 for 3 days. IL-5 and IL-13 levels in the culture supernatants of sorted lung ILC2s were measured by ELISA. **c** Lung ILC2s were sorted from adult offspring of asthmatic or control mothers and transplanted intravenously to *Il7r*-deficient mice. One day after transplantation, IL-33 was administered intranasally to recipient mice. Flow cytometric analysis of the lungs of recipient mice transplanted with sorted

lung ILC2s. In (**a**–**c**), data were pooled from three independent experiments (**a**), each experiment with one or two pregnant dams, or representative of two independent experiments (**b**), each experiment with one pregnant dam per group, or pooled from two independent experiments (**c**) with one pregnant dam per group. Sample sizes were as follows: **a** PBS-PBS: $n = 6$, OVA-PBS: $n = 6$, PBS-HDM: $n = 19$, OVA-HDM: $n = 19$; **b** PBS: $n = 3$, OVA: $n = 3$; **c** PBS: $n = 5$, OVA: $n = 4$. In (**a**, **c**), each dot represents an individual mouse. In (**b**), each dot represents an individual well from an in vitro experiment, with cells sorted and pooled from PBS ($n = 4$) and OVA ($n = 4$) groups of mice. In (**a**–**c**), data are presented as the mean ± SEM. *$p < 0.05$; ns, not significant [unpaired two-tailed Student's $t$-test].

treated with the isotype control (Fig. 3c). IL-7 signaling affects T and B cell development and homeostasis[40], but at least in the lungs of adult offspring, B cells, CD4+ T cells, and Th2 were not reduced (Supplementary Fig. 3c). These suggest that removing fetal lung ILC2s from

asthmatic mothers and resetting the effects of the embryonic period alleviates the enhanced ILC2 response and eosinophilic lung inflammation in adult offspring. Collectively, the results suggest that changes in the lung environment during the embryonic period may contribute

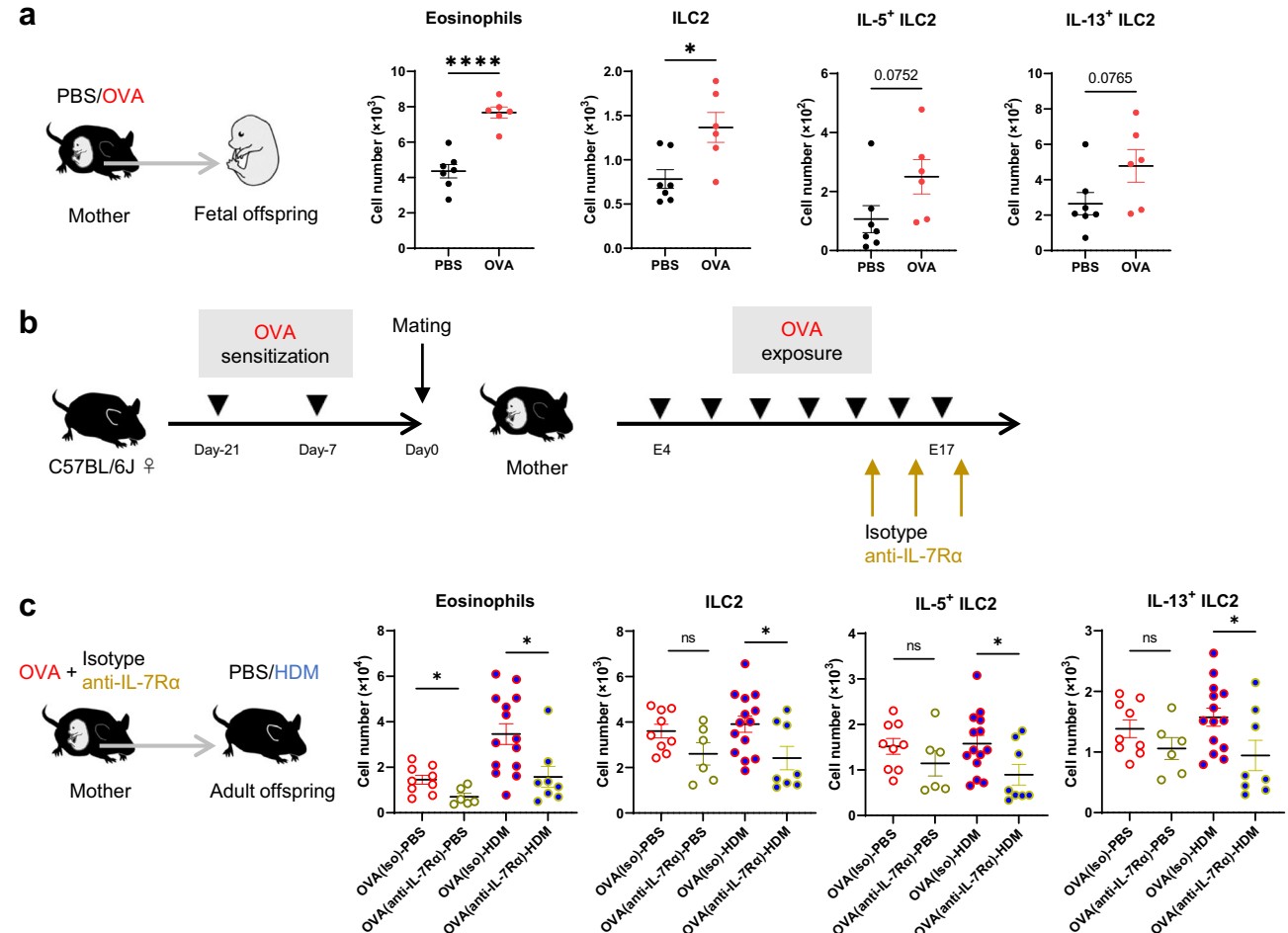

**Fig. 3 | Pulmonary immune changes in embryonic life affect adult allergic responses. a** Flow cytometric analysis of fetal lungs from asthmatic or control mothers (day 18 of fetal life); ILC2s were stimulated with PMA/ionomycin for 4 h, and cytokine production was assessed by intracellular staining. **b** Model of maternal anti-IL-7Rα treatment in late pregnancy. Female mice developed asthma during pregnancy (Fig. 1a) and were treated intravenously with isotype or anti-IL-7Rα antibodies on days 14, 16, and 18 of gestation. **c** Adult offspring of antibody-treated asthmatic mothers received PBS or HDM intranasally (Fig. 1a). Changes in eosinophil and ILC2 cell numbers in the lungs of adult offspring; ILC2s were stimulated with PMA/ionomycin for 4 h, and cytokine production was assessed by intracellular staining. In (**a, c**), data are representative of two independent experiments [(**a**), intracellular staining was evaluated once], each experiment with one pregnant dam per group, or pooled from two independent experiments, each experiment with one or two pregnant dams. Sample sizes were as follows: **a** PBS: $n = 7$, OVA: $n = 6$; **c** OVA(Iso)-PBS: $n = 9$, OVA(anti-IL-7Rα)-PBS: $n = 6$, OVA(Iso)-HDM: $n = 14$, OVA(anti-IL-7Rα)-HDM: $n = 8$. In (**a, c**), each dot represents an individual mouse. In (**a, c**), data are presented as the mean ± SEM. *$p < 0.05$; ****$p < 0.0001$; ns, not significant [unpaired two-tailed Student's $t$-test].

to the enhanced lung ILC2 response and allergen-induced lung inflammation in adult offspring via changes in fetal lung ILC2s.

## Asthma during pregnancy alters the phenotype of ILC2s in the fetal lung

Next, to further investigate the immunological changes, mainly, the alterations of ILC2s, in the fetal lung, ILC2s (CD45⁺, Lin⁻, ST2⁺), immune cells (CD45⁺), and epithelial cells (EpCAM⁺) were sorted from the fetal lung, and single nucleus RNA sequencing (snRNA-seq) analysis was performed (Fig. 4a). Uniform manifold approximation and projection (UMAP) revealed the presence of a variety of cells. We also detected fetal lung ILC2s (Fig. 4b). A volcano plot overview of genes with relatively large fold changes in fetal lung cells from OVA-asthmatic mothers indicated a trend of more upregulated genes in ILC2s than in other immune cells (macrophages) and epithelial cells (type 2 alveolar epithelial cells; AT2) (Supplementary Fig. 4a and Supplementary Data 2). Looking more closely at fetal lung ILC2s, the distribution of UMAP differed between fetal lung ILC2s from OVA-asthmatic and control mothers (Fig. 4c), and many genes were upregulated in fetal lung ILC2s from OVA-asthmatic mothers, with genes involved in

activation and those related to glucocorticoid signaling (Fig. 4d, e)[41–43]. Regarding the ILC2s in the two groups, *Il1rl1*, *Gata3*, and *Nr3c1*, which were upregulated in ILC2s from asthmatic mothers, showed similar changes in protein expression levels, as assessed by flow cytometry (Fig. 4f, g).

Sub-clustering of fetal lung ILC2s resulted in four clusters (Groups 0–3) (Fig. 4h). There was an increased proportion of Group 0 ILC2s in the lungs of fetuses from asthmatic mothers (Fig. 4i), and this cluster tended to be high in genes related to activation markers such as *Il1rl1* and *Gata3*, and genes related to glucocorticoid signaling such as *Nr3c1* (Fig. 4j, k and Supplementary Fig. 4b, c). In addition, scoring with a known gene set for fetal lung ILC2s in the two groups showed increased scores for activation, proliferation, and glucocorticoid signaling in the OVA-asthmatic maternal origin (Supplementary Fig. 4d and Supplementary Data 3)[44,45]. *Nr3c1*, which encodes the glucocorticoid receptor (GR), was most highly expressed in ILC2s, among all the cell types examined in the single-cell analysis of the embryonic lung (Supplementary Fig. 4e). Flow cytometric analysis showed higher levels of NR3C1 protein expression in ILC2s than in non-immune or lineage-positive cells (Supplementary Fig. 4f). Furthermore, the

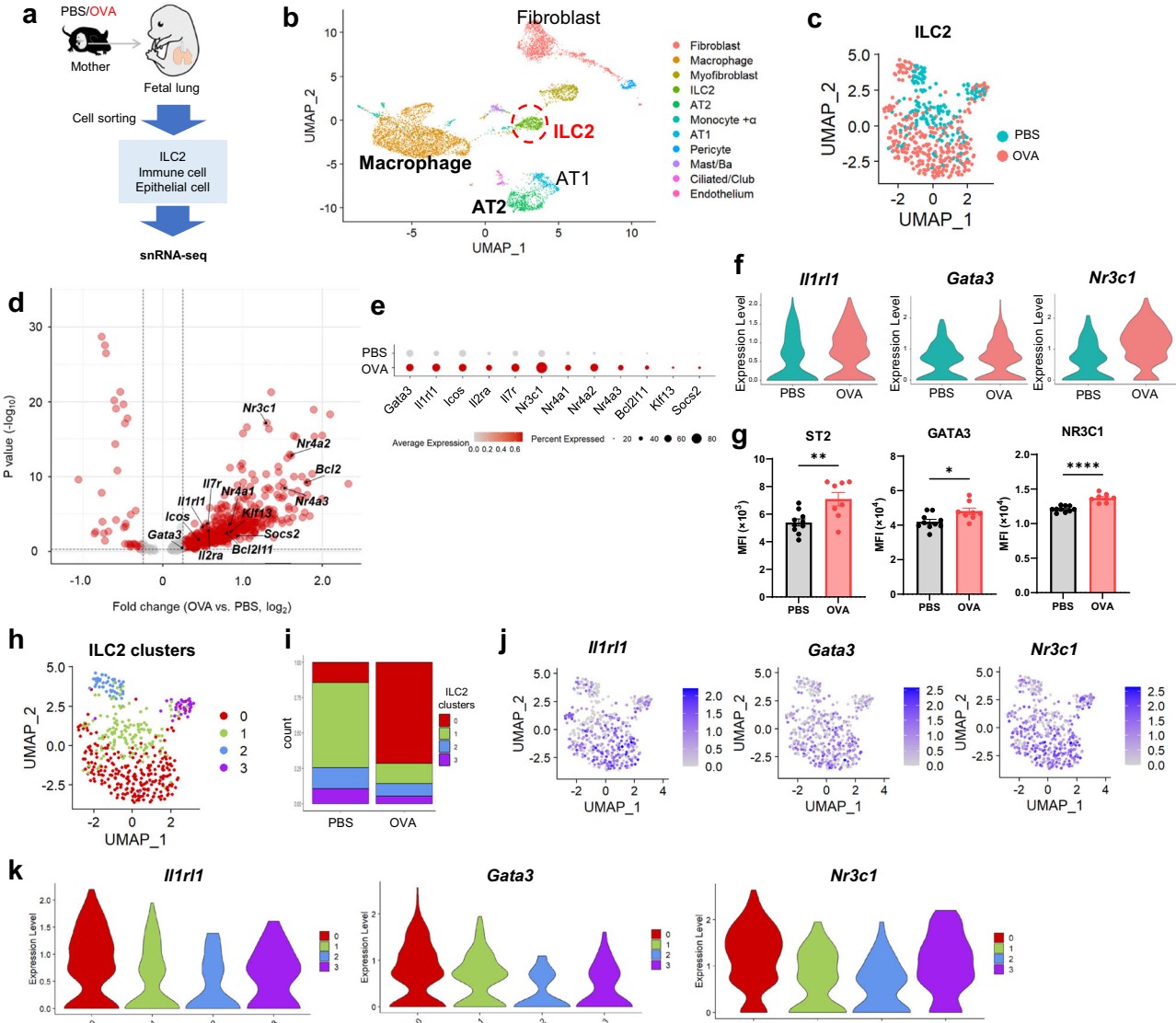

**Fig. 4 | Fetal lung ILC2s from asthmatic mothers exhibit distinct gene expression compared to the controls. a** Experimental design of snRNA-seq in ILC2s, immune cells, and epithelial cells sorted from fetal lungs of asthmatic or control mothers. **b** Uniform manifold approximation and projection (UMAP) visualization of snRNA-seq data from all samples. **c** UMAP of fetal lung ILC2s divided into groups. **d** Volcano plot showing differences in fetal lung ILC2s between the groups. Differentially expressed genes between OVA and PBS were identified using the Find-Markers function from the Seurat package. The Mann-Whitney U test (two-sided) was used to determine statistical significance, with Benjamini-Hochberg correction applied for multiple comparisons. **e** Dot plot of representative genes differentially expressed between the groups. Violin plots showing representative cell surface markers and transcription factors differentially expressed between groups (**f**) and MFI in respective flow cytometry (**g**). **h** UMAP depicting subclusters of ILC2s in the fetal lung. **i** Percentage of ILC2 subclusters compared between groups. Relative expression of *Il1rl1*, *Gata3*, and *Nr3c1* in ILC2s (**j**), and a violin plot displaying expression patterns among ILC2 subclusters (**k**). In (**a–k**), data were derived from one experiment [(**a–f**), and (**h–k**)] with one pregnant dam per group, or two independent experiments (**g**), each experiment with one pregnant dam per group. Sample sizes were as follows: **g** PBS: *n* = 10, OVA: *n* = 8. In (**g**), each dot represents an individual mouse. In (**g**), data are presented as the mean ± SEM. *$p < 0.05$; **$p < 0.01$; ****$p < 0.0001$; ns, not significant [unpaired two-tailed Student's *t*-test in (**a**, **h**)].

greatest change was seen in ILC2s upon the comparison of fetal cells from asthmatic and control mothers (Supplementary Fig. 4g). Thus, asthma during pregnancy upregulates genes related to activation and glucocorticoid signaling in the fetal lung ILC2s.

In addition to these findings in ILC2s, we also observed distinct gene expression changes in other cell types, including macrophages and AT2, in the fetal lung. The data showed that in macrophages, there is an upregulation of genes associated with positive regulation of cell migration and leukocyte differentiation in the fetal lung from asthmatic mothers (Supplementary Fig. 4h). In AT2, we observed an increase in gene expression associated with adherens junction and in utero embryonic development in the fetal lung from asthmatic mothers (Supplementary Fig. 4i). These findings suggest that maternal asthma during the embryonic period affects not only the changes in ILC2s, but also the differentiation and maturation of other immune cells, such as macrophages, and the development of epithelial cells, which are structural elements of the lung.

## Maternal asthma induces epigenetic changes in lung ILC2s in fetuses and adult offspring

To assess whether maternal asthma induces long-term effects on ILC2 responses in the offspring, epigenetic changes in the lung centered on ILC2s were analyzed during fetal and adult life. ILC2s (CD45+, Lin-, ST2+), immune cells (CD45+), epithelial cells (EpCAM+), and fibroblasts (EpCAM-, CD31-, PDGFRβ+) were sorted from the lungs of offspring of OVA-asthmatic and control mothers at both fetal and

adult stages, followed by single-cell assay for transposase-accessible chromatin by sequencing (scATAC-seq) analysis to assess chromatin accessibility (Fig. 5a). UMAP revealed the presence of a variety of cells and identified the target lung ILC2s (Fig. 5b). When the chromatin regions that were more open in the offspring of asthmatic mothers compared to the offspring of control mothers were assessed during fetal and adult life, the proportion of genomic region types was similar (Supplementary Fig. 5a), and the chromatin regions that were more open in the lung ILC2s of asthmatic mothers were partially shared between fetal and adult offspring (Fig. 5c). Analysis of the enriched regions in fetal lung ILC2s from asthmatic mothers showed the enrichment of a group of genes involved in cell development (Supplementary Fig. 5b). Furthermore, a group of genes involved in cell migration and MAPK signaling pathway were enriched in adult offspring lung ILC2s from asthmatic mothers (Supplementary Fig. 5c). The open chromatin regions shared between the fetus and adult offspring showed enrichment of a set of genes involved in the MAPK signaling pathway (Fig. 5d). The MAPK signaling activated through GATA3 phosphorylation is an important pathway in ILC2s (Fig. 5e)[46,47]. In lung ILC2s from adult offspring of asthmatic mothers, peak changes were observed in *Mapkap3* and *Map2k3*, genes involved in MAPK signaling, and *Gata3*, a master transcription factor, some of which were shared with changes in fetal lung ILC2s (Fig. 5f and Supplementary Fig. 5d). These regions overlapped with the peak in ChIP-seq of H3K4me3 for adult lung ILC2s, suggesting that they are involved in transcriptional activity. Based on the results of the scATAC-seq, we investigated the activation of the MAPK pathway. In adult offspring from asthmatic mothers, we observed increased phosphorylation of p38 in lung ILC2s at baseline. Furthermore, after in vivo stimulation with IL-33, the phosphorylation of p38 was significantly increased compared to controls (Supplementary Fig. 5e). In addition, GR-binding sites were identified in regions of increased accessibility with *Gata3*, the master transcription factor for ILC2s (Fig. 5f). The function of this binding site for *Gata3* is not clear. However, it has been reported that CD4[+] T cells in *Nr3c1*[(-/-)] have a reduced capacity to produce type 2 cytokines[41], and that glucocorticoid exposure can enhance ILC2 function[48], suggesting that glucocorticoid signaling via GR may be involved in regulating gene expression in *Gata3* to activate ILC2s. To validate the function of GR in ILC2s, we analyzed mice deficient in GR in ILC2s (*Nr3c1*[fl/fl] *Il7r*[cre/+]). GATA3 expression was impaired in lung ILC2s of these mice, indicating that GR regulation is indeed important for GATA3 expression (Fig. 5g). These suggest that maternal asthma can have long-lasting effects on fetal lung ILC2s, leading to epigenetic changes in adult offspring. Additionally, glucocorticoid signaling may play a role in this process. In addition, to verify whether the changes in ILC2 in the offspring of asthmatic mothers are specific to the Alum-induced asthma model, we induced asthma in the mothers using *Alternaria*, a model of asthma caused by mucosal sensitization (Supplementary Fig. 6a). Fetuses of *Alternaria* asthmatic mothers showed increased proportions of eosinophils and ILC2s in the lung compared to controls, exhibiting changes similar to those observed in the OVA asthma model (Supplementary Fig. 6b). NR3C1 levels were significantly elevated in fetal lung ILC2s, and a slightly higher GATA3 expression was also observed (Supplementary Fig. 6b). We further sorted these fetal lung ILC2s for bulk ATAC-seq analysis and found an increased open chromatin region at the *Gata3* gene locus in fetal lung ILC2s from *Alternaria* asthmatic mothers (Supplementary Fig. 6c). Additionally, we observed an increase in the numbers of lung eosinophils and the proportion of cytokine-producing ILC2s in adult offspring of *Alternaria* asthmatic mothers after intranasal administration of mite antigen (Supplementary Fig. 6d). These results suggest that maternal asthma has a similar effect on offspring, even in different models of maternal asthma.

## Effects of glucocorticoid exposure during pregnancy on lung ILC2s in offspring are similar to those observed during maternal asthma

As asthma during pregnancy alters the ILC2 phenotype of the fetal lung, we explored potential mechanisms acting *in utero*. GR, encoded by *Nr3c1*, binds glucocorticoids and regulates gene expression[49]. Given the enhancement of glucocorticoid signaling and increase in *Nr3c1* gene expression in the fetal lung ILC2s of OVA-asthmatic mothers, we investigated glucocorticoid levels in serum and amniotic fluid. The level of corticosterone, a glucocorticoid and stress hormone, was elevated in fetal amniotic fluid and fetal serum from OVA-asthmatic mothers at 16 days of gestation (Fig. 6a), and a similar trend was observed in maternal serum (Supplementary Fig. 7a). Serum corticosterone levels increase during pregnancy[50], and OVA-asthmatic mothers tended to have even higher serum corticosterone levels than control mothers, especially from around 12 days of gestation (Supplementary Fig. 7b). As glucocorticoid exposure during pregnancy has been reported to lead to increased allergen-induced lung inflammation in the offspring[51,52], we used a model mimicking maternal stress using a synthetic glucocorticoid dexamethasone (DEX). DEX was administered orally to pregnant mice from 12 days of gestation, and the effects on the lungs of the offspring were examined (Fig. 6b). Eosinophil numbers were elevated in the fetal lungs of DEX-treated mothers (Fig. 6c), similar to those of asthmatic mothers, suggesting an altered immune environment in the fetal lungs due to DEX. Single-cell analysis was then performed to assess the changes in fetal lung ILC2s derived from DEX-treated mothers (Fig. 6d). Fetal lung ILC2s from DEX-treated mothers showed a different distribution of UMAPs compared with that for the control mothers, with increased expression levels of the master transcription factor *Gata3* and glucocorticoid receptor *Nr3c1*; similar changes were also observed in the protein expression levels (Fig. 6d–f). We created a gene set that was elevated in fetal lung ILC2s from DEX-treated mothers (Supplementary Data 3). Scoring the fetal lung ILC2s in an asthmatic mother model based on this gene set showed higher scores in fetal lung ILC2s from asthmatic mothers (Fig. 6g). Finally, when we assessed changes in adult offspring in the DEX-treated maternal model, we found increased severity of mite antigen-induced eosinophilic pneumonia in adult offspring from DEX-treated mothers, with increased lung ILC2 numbers and reactivity (Fig. 6h). These suggest that changes in pulmonary ILC2s in offspring due to chronic glucocorticoid elevation during pregnancy overlap with changes in lung ILC2s in offspring from asthma models and contribute to allergen-induced lung inflammation in adult offspring. To further investigate the synergistic effect of allergen exposure and corticosteroid treatment during pregnancy, we conducted an experiment where both were applied to the maternal model. We induced chronic asthma in pregnant mice and concurrently administered oral DEX from E12 onwards (Supplementary Fig. 7c). The data showed that there were no significant differences in lung eosinophil numbers between the adult offspring from mothers in the DEX combination group and those in the allergen-only group (Supplementary Fig. 7d). While there were no major changes in cytokine-producing ILC2s, there was a tendency for an increased number of ILC2s in the adult offspring of mothers receiving DEX (Supplementary Fig. 7d). Additionally, in another experiment using non-pregnant mice to assess the effects of DEX on our OVA/Alum asthma model, we found no significant differences in eosinophilic inflammation in the lungs between the two groups (Supplementary Fig. 7e). These results suggest that, within the context of our experimental model, DEX treatment did not alleviate lung inflammation in the mothers, which likely reflects the lack of changes in eosinophil numbers in their adult offspring. However, we observed a trend towards an increase

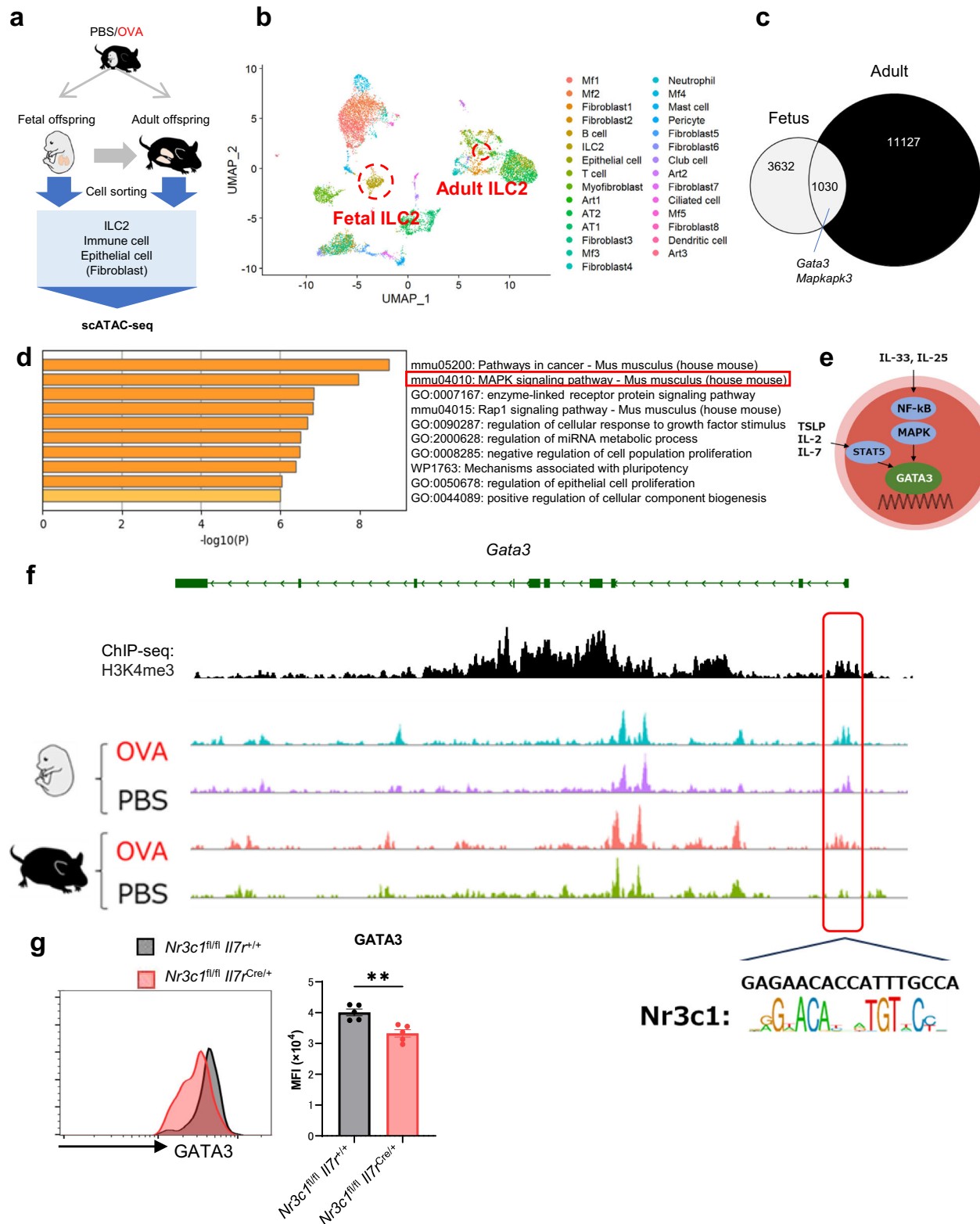

in lung ILC2 numbers in the offspring of mothers treated with DEX, suggesting that systemic administration of steroids to asthmatic mothers may synergistically influence lung ILC2s in their adult offspring alongside the effects of asthma itself.

## Discussion

Our studies show that maternal asthma has long-term effects that alter fetal lung ILC2s and lead to increased lung ILC2 responsiveness in adults and that the mechanism involves epigenetic changes and glucocorticoid signaling.

First, our study showed that adult offspring of asthmatic mothers exhibit increased allergen-induced lung inflammation, independent of the maternal asthma antigens. As mentioned above, there are conflicting reports as to whether antigen identity between mother and offspring is necessary[15,16,19,20,22]. Though both adaptive and innate immunity can be involved in the intergenerational transmission of

**Fig. 5 | Epigenetic changes in lung ILC2s of asthmatic mothers' offspring overlap from embryonic to adult stages. a** Experimental design for scATAC-seq. ILC2s, immune cells, and epithelial cells were sorted from fetal lungs of asthmatic or control mothers, and ILC2s, immune cells, epithelial cells, and fibroblasts were sorted from adult offspring lungs of asthmatic or control mothers. **b** UMAP visualization of scATAC-seq data from all samples. **c** Number of differentially accessible peaks and overlapping regions in lung ILC2s of fetal and adult offspring of asthmatic mothers (pct > 0.05, log2FC < 0.1, $p < 0.5$). Differentially accessible peaks between OVA and PBS were identified using the FindMarkers function from the Signac package with logistic regression (LR) as the statistical test. The Mann-Whitney U test (two-sided) was used for significance testing, and Benjamini-Hochberg correction was applied for multiple comparisons. **d** Enrichment analysis of genes associated with overlapping regions that were more open in lung ILC2s of

fetal and adult offspring from asthmatic mothers (top 10 positions shown). Gene set enrichment analysis was performed using Metascape. Statistical significance was determined using the Fisher's exact test (two-sided), and Benjamini-Hochberg correction was applied for multiple comparisons. **e** Simplified diagram showing intracellular signaling of ILC2s. **f** scATAC-seq and ChIP-seq results in lung ILC2s of offspring from asthmatic or control mothers in the *Gata3* locus. **g** GATA3 expression levels in lung ILC2s of $Nr3c1^{fl/fl} Il7r^{+/+}$ mice and $Nr3c1^{fl/fl} Il7r^{cre/+}$ mice, shown as MFI bar graphs and histograms. In (**a–f**), data were derived from one experiment with one pregnant dam per group. In (**g**), data were from one experiment, with each dot representing an individual mouse. Sample sizes were as follows: **g** $Nr3c1^{fl/fl} Il7r^{+/+}$: $n = 5$, $Nr3c1^{fl/fl} Il7r^{cre/+}$: $n = 5$. In (**g**), data are presented as the mean ± SEM. **$p < 0.01$ [unpaired two-tailed Student's $t$-test in (**g**)].

asthma, looking back at previous reports closely, in a model that included not only antigen sensitization during pregnancy but also antigen exposure in the airways, adult offspring of asthmatic mothers worsened asthma in an antigen-non-specific manner, regardless of whether the difference was significant[15,16,22]. This suggests that the mechanism of antigen-non-specific transmission of asthma susceptibility between mother and child requires a maternal burden, such as eosinophilic pneumonia.

We have shown that asthma during pregnancy enhances lung ILC2 responses in adult offspring, suggesting that the mechanism of transmission of asthma susceptibility involves immune cell programming under the influence of changes in the embryonic environment. Previous studies have reported that the adoptive transfer of splenic dendritic cells from the offspring of asthmatic mothers induces a more Th2 phenotype compared to controls[21], supporting the possibility that maternal asthma may alter the nature of immune cells in the offspring. Although we focused on changes in ILC2s in the present study, further studies are needed to understand how these altered lung ILC2s engage in cross-talks with surrounding cells to worsen allergen-induced lung inflammation, as maternal asthma can also affect other immune cells, airway epithelial cells and peripheral nerves associated with the airways in the offspring[20,21,53].

There are only a few studies on the effects of maternal asthma on fetal ILC2s. Further, there are reports of an association between phenotypic changes in cord blood ILC2s in asthmatic mothers and lung function in their children[54]. Although our study focused on changes in the fetal lung, the finding that ILC2 priming early in life can affect and activate lung ILC2s in individuals in the long term has been investigated mainly for lung ILC2s during the neonatal period. Neonatal lung ILC2s are activated by endogenous IL-33, the effects of which are related to adult lung ILC2 function[55]. Antimicrobial treatment during pregnancy has a long-term effect on increased allergic response in adulthood by causing epigenetic changes in neonatal pulmonary ILC2s via breastfeeding[56]. Given that adult lung ILC2s have three origins, embryonic, neonatal, and adult, it is not surprising that the long-term effects on lung ILC2s observed in the neonatal period can also be monitored for lung ILC2s in embryonic life. However, this study did not investigate the actual identification and functional elucidation of lung ILC2 subsets of embryonic origin in adult offspring.

In our study, snRNA-seq analysis of fetal lung ILC2 in asthmatic mothers showed increased gene expression associated with activation and glucocorticoid signaling. At the embryonic stage, lung ILC2s can receive IL-33 stimulation and activate and produce cytokines, at least in the presence of IL-7[57]. The increase in the expression of activation markers in ILC2s in the fetal lungs of asthmatic mothers is interesting from the point of view of "trained immunity," as it is thought that they have experienced some kind of stimulus during embryonic development. However, further research is needed into the factors involved. On the other hand, it has been reported that ILC2 numbers are enhanced by glucocorticoid exposure[48], suggesting that lung ILC2s may be primed at the embryonic stage by glucocorticoid signaling. The

highest expression of glucocorticoid receptors in ILC2s compared to epithelial cells and other immune cells of the cell types studied, and the greatest change compared to controls, also supports that the effects of glucocorticoids are more accentuated in ILC2s.

Detailed scATAC-seq analysis of fetal and adult lung ILC2s in the offspring of asthmatic mothers showed that there was a partial overlap of differential accessibility sites between them, suggesting that the embryonic environment causes epigenetic changes in embryonic lung ILC2s, which are retained during development into adult lung ILC2s. We performed enrichment analysis on the differential accessibility sites regions shared between fetal and adult lung ILC2s in the offspring of asthmatic mothers, revealing an enrichment of gene sets associated with the MAPK signaling pathway. In ILC2s, the MAPK signaling pathway is involved in the intracellular signaling of activation in response to the stimulation of IL-33[33,46,47]. Previous reports have shown that epigenetic regulation of MAPK-related genes themselves is associated with activation of intracellular signaling[58], suggesting that chromatin accessibility may be involved in GATA3 phosphorylation and increased cytokine production through similar changes in ILC2s. In our study, lung ILC2s from adult offspring of asthmatic mothers showed enhanced cytokine production in in-vitro culture in the presence of IL-33, suggesting that a signaling response pathway to IL-33 stimulation is involved in the enhancement of adult lung ILC2 function.

In addition, the *Gata3* locus, the master transcription factor of ILC2s, also contained a region of increased chromatin accessibility that was common to both fetal and adult lung ILC2s from asthmatic mothers. This open chromatin region was also detected as a peak in the ChIP-seq of H3K4me3 performed simultaneously with ILC2s, suggesting that it is involved in transcriptional activation of the gene. Furthermore, it is interesting to note that a GR motif is detected in this peak. In general, the action of glucocorticoids has been emphasized as immunosuppression, with known negative regulatory effects, such as the increased expression of apoptotic genes in T cells and decreased expression of pro-inflammatory cytokines[59]. However, the effects of glucocorticoids on the immune system are complex and not fully understood, with some reports suggesting that they are important, for example, in maintaining CD8 T-cell and CD4 T-cell function[41]. In addition, previous reports have shown that glucocorticoid exposure can paradoxically increase Th2 cytokine production[60–63], and that GR reduces Th1 cytokine production but increases Th2 cytokine production in an experiment using mice lacking the CD4 T-cell specific *Nr3c1*[41]. As mentioned above, there are reports that glucocorticoid exposure increases ILC2 function[48]. The role of GR in ILC2s requires further study. Nevertheless, our study results showed that maternal glucocorticoid treatment increased Gata3 expression in fetal lung ILC2s, and it is possible that glucocorticoid signaling may situationally enhance Th2 cytokine production in ILC2s. It has been reported that maternal glucocorticoid administration causes a reduction in basal glucocorticoid levels in the offspring, leading to reduced T-bet expression in CD8 T cells and loss of accessibility to the GR-binding site at the *Tbx21* locus[64]. It suggests that in lymphocytes, the effects of GR on the key

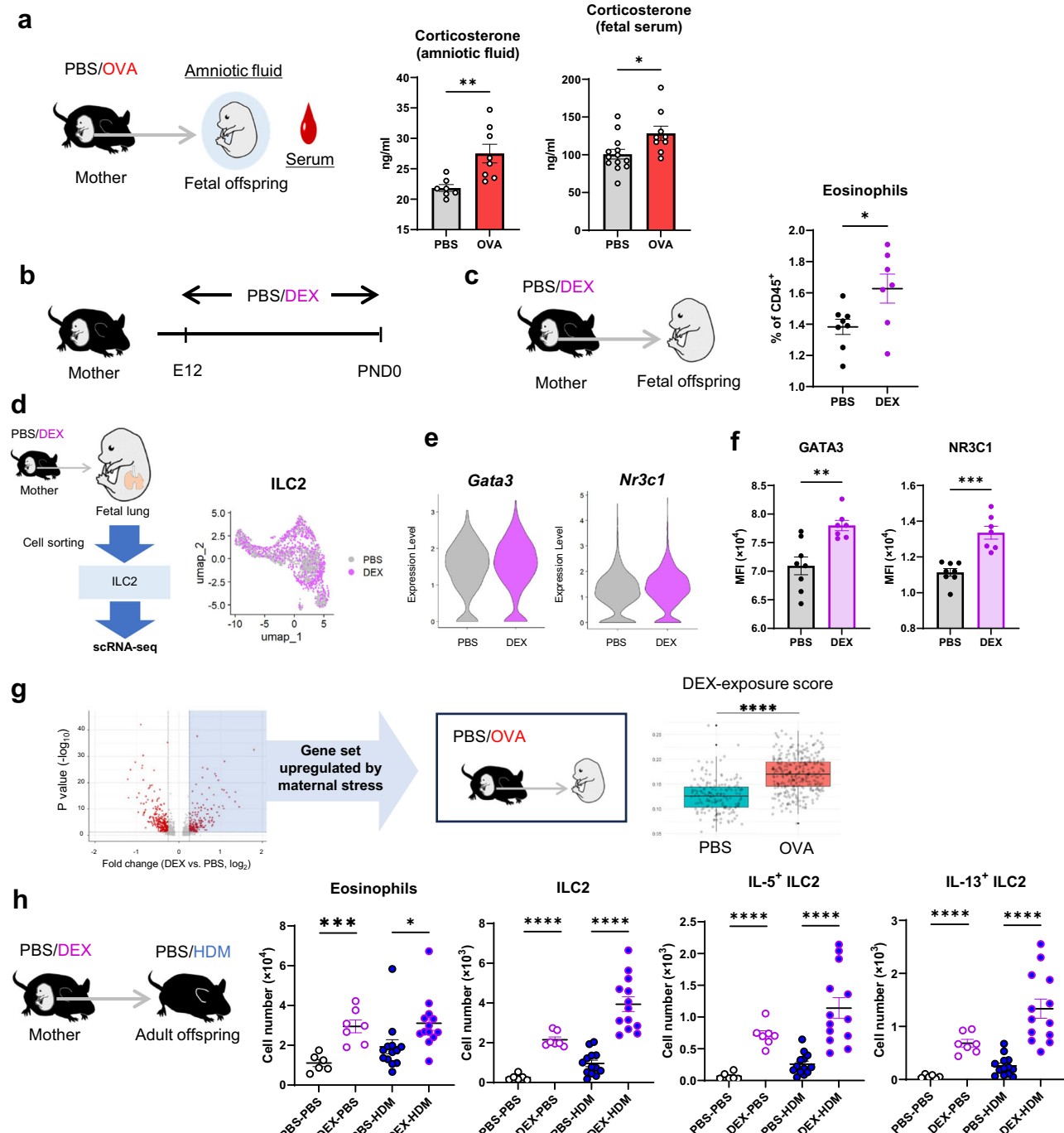

**Fig. 6 | Glucocorticoid signaling contributes to asthma susceptibility transmission. a** Concentrations of corticosterone in amniotic fluid and serum of fetuses from asthmatic or control mothers. **b** Model of chronic glucocorticoid treatment during pregnancy. Female mice were treated orally with water containing PBS or dexamethasone (DEX) from day 12 of gestation until the end of pregnancy. **c** Percentage of eosinophils in the lungs of fetuses from DEX or control mothers (day 18 of gestation). **d** Left: Experimental design of scRNA-seq in ILC2s sorted from fetal lungs derived from DEX or control mothers. Right: UMAP of fetal lung ILC2s divided from the different groups. Violin plots showing representative transcription factors differentially expressed between groups (**e**) and bar plots of their respective MFI by flow cytometry (**f**). **g** The volcano plot on the left shows expression changes in fetal lung ILC2s from DEX-treated and control mothers. The box plot on the right shows fetal lung ILC2s in the asthma model, scored using the set of genes that were upregulated in fetal lung ILC2s from DEX-treated mothers and compared between groups. **h** Changes in eosinophil and ILC2 counts in the lungs of adult offspring from DEX or control mothers; ILC2s were stimulated with PMA/ionomycin for 4 h, and cytokine production was assessed by intracellular staining. In (**a–h**), data are representative of two experiments [(**a, c, f, h**), fetal serum was pooled in (**a**)], each experiment with one to three pregnant dams per group, or one independent experiment [(**d–e**) and (**g**)] with one pregnant dam per group. Sample sizes were as follows: **a** Corticosterone (amniotic fluid): PBS: $n = 7$, OVA: $n = 8$; Corticosterone (fetal serum): PBS: $n = 13$, OVA: $n = 9$; **c, f** PBS: $n = 8$, DEX: $n = 7$; (**h**) PBS-PBS: $n = 6$, DEX-PBS: $n = 7$, PBS-HDM: $n = 13$, DEX-HDM: $n = 13$. In (**a, c, f, h**), each dot represents an individual mouse. In (**a, c, f, h**), data are presented as the mean ± SEM. For box plots in (**g**): Data are displayed as min to max, showing all points. The midline represents the median, the box indicates the interquartile range (IQR) between the 25th (Q1) and 75th (Q3) percentiles, and the whiskers extend to the minimum and maximum values within 1.5 times the IQR. *$p < 0.05$; **$p < 0.01$; ***$p < 0.001$; ****$p < 0.0001$ [unpaired two-tailed Student's $t$-test in (**a, c, f, h**), and two-tailed Wilcoxon rank sum test in (**g**)].

transcription factors themselves are important for maintaining effector function. In our study, a GR-binding site was found in the open chromatin region of the *Gata3* locus, and this state of increased accessibility for GR binding may be involved in the enhancement of the functions of ILC2s from offspring of asthmatic mothers.

Maternal glucocorticoid treatment enhancing ILC2 function in offspring is clinically significant, especially as many asthmatic mothers receive steroid therapy during pregnancy. Human studies suggest that managing asthma during pregnancy reduces the risk of asthma in children[18], while other reports suggest that steroid use during pregnancy may increase the risk of asthma in offspring[65]. In our study, DEX treatment in models of asthmatic mothers showed a trend towards increased ILC2s in the lungs of the offspring, suggesting that maternal glucocorticoid use in asthmatic mothers may enhance ILC2 function in the offspring and increase their risk of asthma. In our model, DEX treatment did not sufficiently alleviate maternal eosinophilic inflammation, limiting our ability to assess the combined effects of asthma symptom improvement with steroid therapy and the effects of steroid administration itself. Further research is needed to understand the long-term effects of steroid therapy for asthma during pregnancy on offspring.

Overall, this study identified a link between fetal lung ILC2 numbers, which are altered by the embryonic environment, and increased lung ILC2 responsiveness in adulthood from an epigenetic perspective. The fact that glucocorticoids are partly responsible for the transmission of asthma susceptibility in maternal asthma provides us with a more general and common mechanism whereby maternal conditions affect offspring.

## Methods

### Mice
Wild-type C57BL/6J mice were purchased from KBT Oriental (Breeder: Jackson Laboratory Japan). *Il7r*-Cre mice were provided by Professor Hans-Reimer Rodewald. GR-flox (*Nr3c1*-flox) mice were provided by Professor Koichi Ikuta. All mice were housed in the specific pathogen-free animal facility at Kyushu University with a 12-h light cycle. Both genders were used. All experimental procedures were approved with a No. A23-234-1 and A24-201-0 by the Kyushu University Animal Experiment Committee, and the care of the animals was in accordance with institutional guidelines. All experiments were performed to keep animal pain, stress, and numbers to a minimum.

### OVA and house dust mite model
Female mice aged 6–8 weeks were sensitized by intraperitoneal administration of 200 μL PBS/alum (Thermo, Imject Alum, 77161) containing alum (25 μL) or 200 μL PBS/alum containing OVA (100 μg) (Sigma, A5503) and alum (25 μL) on day -21 and day -7. After the second sensitization, sensitized female mice were bred to untreated male mice. Pregnant mice were given OVA (20 μL PBS containing 500 μg OVA) or PBS intranasally every 2–3 days from day 4 to day 17 of gestation. Analysis of offspring was performed during pregnancy or analysis of offspring after birth. For fetal analysis, the mother was sacrificed, and her uterus, blood, and lungs were collected on gestation days 16 and 18. In addition, on day 16 of gestation, amniotic fluid was collected from the uterus and fetal blood was collected for analysis. Fetal lungs were collected for analysis on day 18 of gestation. For the analysis of adult offspring, HDM (20 μL PBS containing 10 μg HDM) or PBS was administered intranasally to 7–9 week old offspring for five consecutive days. The day after final challenge, the mice were sacrificed, and organs were dissected for analysis. BAL infiltrates were collected by 3 consecutive flushes of the airways with 1 mL of PBS. Lung and blood were also collected.

In the model of adult offspring sensitized and exposed to HDM from asthmatic or control mothers: Adult offspring from asthmatic mothers were sensitized with HDM (20 μL PBS containing 1 μg HDM) on day 0 and subsequently exposed to HDM (20 μL PBS containing 10 μg HDM) intranasally from days 7 to 11. The day after final challenge, the mice were sacrificed, and the lungs were dissected for analysis.

### Measurement of airway hyper-responsiveness
Airway responsiveness was measured according to a protocol described in previous reports[66]. Mice were anesthetized intraperitoneally with a mixture of ketamine and sodium pentobarbital, and the trachea was cannulated through a tracheotomy. After paralysis, mice were mechanically ventilated (tidal volume 0.3 mL, respiratory rate 120 breaths/min). The airway opening pressure was measured with a differential pressure transducer and recorded continuously. Acetylcholine (Sigma, 60-31-1) doses were increased stepwise (1.25–80 mg/mL) for 1 min using an ultrasonic nebulizer (NE-U07; Omron Corporation, Kyoto, Japan). Data were expressed as evoked concentration 150 (PC150), the concentration at which airway pressure was 150% of baseline values; PC150 was calculated by log-linear interpolation for individual animals; PC150 values were expressed as log (100 × PC150).

### Lung histology
Lung tissues were fixed in 4% paraformaldehyde for 24 h, and sections were prepared and stained with H&E to assess inflammation. To grade inflammatory cell infiltration, peribronchial cell counts were determined using a 5-point grading system as follows: 0, normal; 1, few cells; 2, a ring of inflammatory cells one cell layer deep; 3, a ring of inflammatory cells of 2–4 cells deep; and 4, a ring of inflammatory cells of more than 4 cells deep[67]. Ten fields were counted randomly from each section, and the mean score per mouse was calculated, with the scoring performed in a blinded manner.

### Lung homogenate preparation
Post-sacrifice, the right lung lobes were homogenized in 1 mL of PBS using a Micro Smash™ MS-100 (TOMY). The lung homogenate was then centrifuged at 9000 × g for 10 min at 4 °C. The supernatant was collected and stored at −80 °C for further analysis.

### IL-7Ra antibody administration
For depletion of fetal ILC2, IL-7Ra blockade was performed by intravenous injection of 200 μg isotype control (Bio X Cell, BE0089) or anti-IL-7Rα antibody (Bio X Cell, BE0065) into pregnant female mice on days 14, 16, and 18 of gestation. IL-7Ra blockade in non-pregnant asthmatic mice was performed over the same time course as in pregnant mice on days 14, 16, and 18, with the first OVA nasal drop defined as day 4.

### Alternaria asthma model during pregnancy
Pregnant female mice were treated with intranasal administration of *Alternaria* (20 μL PBS containing 25 μg *Alternaria alternata* extract (Greer Laboratories, XPM1D3A2.5)) or PBS every 3 days from gestation day 0 to day 18. For fetal analysis, pregnant mice were sacrificed on gestation day 18, and fetal lungs were collected for assessment. For the analysis of adult offspring, HDM (20 μL PBS containing 10 μg HDM) or PBS was administered intranasally to 7–9 week old offspring for five consecutive days. The day after final challenge, the mice were sacrificed, and the lungs were collected for detailed analysis.

### Maternal stress mimetic model with DEX
For DEX administration, untreated 9-week-old pregnant mice were obtained and DEX was administered in drinking water from day 12 of gestation (DEX was dissolved in PBS to a concentration of 10 mg/ml and further diluted to 1.3 μg/ml in drinking water). The dose and timing of administration were adopted and modified from studies in which DEX was administered to pregnant mice; DEX was treated with drinking water to minimize unnecessary handling and injection stress. Once parturition was confirmed, the water was changed to normal drinking

water. The analysis of the offspring was the same as for the maternal OVA models.

## Combination model of OVA asthma and DEX treatment during pregnancy

Pregnant mice in the maternal OVA asthma model described in the Methods section were given PBS or DEX in the drinking water from day 12 of gestation. This method of DEX administration follows the protocol described in the "Maternal stress mimetic model with DEX" section of the Methods. The analysis of adult offspring was performed in the same way as for the maternal OVA models.

## Preparation of single-cell suspensions from lung tissue

For the preparation of lung immune cells for flow cytometry analysis and scRNA-seq, lung tissue was digested in RPMI enzyme solution containing collagenase D (2 mg/ml, Roche, 11088882001) and DNase I (0.1 mg/ml, Roche, 10104159001) at 37 °C for 60 min (for adults) or 40 min (for fetuses). Digestion was performed by changing the solution every 20 min.

For the preparation of lung immune cells, fibroblast, and epithelial cells for scATAC-seq, lung tissues were digested in RPMI enzyme solution containing collagenase D (2 mg/ml), DNase I (0.1 mg/ml), and Dispase II (0.2 mg/ml, Roche 04942078001) at 37 °C for 40 min (for adults) or 20 min (for fetuses). Digestion was performed by changing the solution every 20 min. And then, lung tissues were digested in RPMI enzyme solution containing collagenase D (2 mg/ml), DNase I (0.1 mg/ml), Dispase II (0.2 mg/ml) and Liberase TH (0.01 mg/ml, Roche 5401151001) at 37 °C for 20 min.

The digested samples were filtered through 70 μm strainers and suspended in 36% Percoll (Cytiva, 17-0891-01) diluted with RPMI, 72% Percoll diluted with PBS was injected into the bottom and centrifuged at $800 \times g$ for 20 min at 25 °C. Cells were collected in the interlayer between 36% and 72% Percoll, washed with RPMI supplemented with FBS (2% v/v), and stained for flow cytometry.

## Flow cytometry analysis

Cells were stained on ice in a V-bottomed 96-well plate. 123count eBeads Counting Beads (50 μL, Thermo Fisher Scientific, 01-1234-42) was used to calculate cell numbers.

For surface marker staining, cells were incubated for 5 min on ice with CD16/CD32 Monoclonal Antibody (1:200, Thermo Fisher Scientific, 14-0161-86) diluted with staining buffer (PBS supplemented with 0.5% (v/v) BSA, 2 mM EDTA). Cells were then stained for surface antigens for 15 min on ice in the presence of Fixable Viability Dye eFluor 780 (1:1000, Thermo Fisher Scientific, 65-0865-14), and fixed for 30 min on ice with eBioscience IC Fixation Buffer (Thermo Fisher Scientific, 00-8222-49) as per manufacturer's instruction. The cells were resuspended in staining buffer.

For transcriptional factor staining, cells were incubated for 5 min on ice with CD16/CD32 Monoclonal Antibody (1:200, Thermo Fisher Scientific, 14-0161-86) diluted with staining buffer. Cells were then stained for surface antigens for 15 min on ice in the presence of Fixable Viability Dye eFluor 780 (1:1000). Intracellular staining was performed with eBioscience Foxp3/Transcription Factor Staining Buffer Set (Thermo Fisher Scientific, 00-5523-00) as per manufacturer's instructions. In brief, cells were fixed with Foxp3 Fixation/Permeabilization working solution (Foxp3 Fixation/Permeabilization Concentrate: Foxp3 Fixation/Permeabilization Diluent = 1: 3) for 30 min on ice, and wash with 1X Permeabilization Buffer. The cells were stained for intracellular antigens in 1X Permeabilization Buffer o/n at 4 °C. The cells were resuspended in staining buffer.

For cytokine staining, cells were stimulated with Phorbol 12-myristate 13-acetate (PMA) (30 ng/ml, Sigma, P8139), ionomycin (0.5 μg/ml, Sigma, 10634), and Brefeldin A Solution (BFA) (1:1000, Invitrogen, 00-4506-51) for 4 h at 37 °C in 5% CO2 incubator. And then,

cells were incubated for 5 min on ice with CD16/CD32 Monoclonal Antibody (1:200) diluted with staining buffer. Cells were then stained for surface antigens for 15 min on ice in the presence of Fixable Viability Dye eFluor 780 (1:1000). Intracellular staining was performed with eBioscience Foxp3/Transcription Factor Staining Buffer Set as per manufacturer's instructions. The cells were resuspended in staining buffer.

For analysis of phospho-p38, cells were incubated on ice for 5 min with CD16/CD32 Monoclonal Antibody (1:200, Thermo Fisher Scientific, 14-0161-86) diluted in staining buffer. Surface antigens were stained for 15 min on ice in the presence of Fixable Viability Dye eFluor 780 (1:1000). Cells were then fixed by incubating with BD Phosflow Lyse/Fix Buffer (BD Biosciences, 558049) at 37 °C for 10 min. Permeabilization was performed using BD Phosflow Perm Buffer III (BD Biosciences, 558050), which had been stored at −20 °C, with cells incubated at 4 °C for 30 min. Following permeabilization, cells were washed twice with staining buffer and then incubated at 4 °C overnight in staining buffer with AlexaFluor647-conjugated anti-p-p38 (1:50, BD Biosciences, 612595) for intracellular staining. The cells were resuspended in staining buffer.

Flow cytometry analysis was performed by LSRFortessa (BD Biosciences), and data were analyzed with FlowJo (version 10, BD Biosciences).

For cell sort, cells were incubated for 5 min on ice with CD16/CD32 Monoclonal Antibody (1:200) diluted with staining buffer. Cells were then stained for surface antigens for 15 min on ice in the presence of Fixable Viability Dye eFluor 780 (1:1000). The cells were resuspended in staining buffer. The cells were sorted by FACSMelody (BD Biosciences), and data were analyzed with FlowJo.

Gating strategies and antibodies for all experiments were shown in Supplementary Fig. 9 and Supplementary Data 4.

## In vitro cultures of ILC2

ILC2s were sorted from lungs of adult offspring of OVA- or PBS- treated mothers. All cell cultures were performed in RPMI supplemented with penicillin and streptomycin, 2-mercaptoethanol, and 10% FBS. For ILC2 activation, ILC2s (3,000 cells) in 200 μL of culture medium were stimulated with IL-33 (10 ng/ml) (R&D, 3626-ML-010/CF(519-54231)) in 96-well polystyrene round-bottom plate. After 3 days, supernatants were harvested and measured IL-5 and IL-13 by ELISA. The cells were stained for flow cytometry analysis.

## ILC2 transplantation

ILC2s were sorted from lungs of adult offspring of OVA- or PBS- treated mothers. Five thousand purified ILC2s were injected intravenously into each *Il7r*-deficient mouse, whose CD4 and CD8A blockade was achieved by intraperitoneal injection of 250 μg anti-CD4 antibody (Bio X Cell, GK1.5, BP0003-1) and anti-CD8A (Bio X Cell, 2.43, BP0061) antibody 1 day prior to transplantation. One day after cell transfer, the mice received a single intranasal injection of IL-33 (0.25 μg) (PeproTech, 210-33) and lung ILC2s were analyzed by flow cytometry the next day.

## ELISA

The levels of corticosterone in serums or amniotic fluids were measured by DetextX Corticosterone ELISA Kit (Arbor Assays, K014-H1) according to the manufacturer's instructions. The levels of IL-5 and IL-13 in the cultured cell supernatants were measured by Mouse IL-5 Uncoated ELISA Kit (Invitrogen, 88-7054-88), Mouse IL-13 Uncoated ELISA Kit (Invitrogen, 88-7137-88) according to the manufacturer's instructions. The levels of IL-33, TSLP, and IL-25 in the lung homogenate supernatants were measured by Mouse IL-33 Uncoated ELISA Kit (Invitrogen, 88-7333-88), Mouse TSLP Uncoated ELISA Kit (Invitrogen, 88-7490-88), Mouse IL-17E/IL-25 Uncoated ELISA Kit (Invitrogen, 88-7002-88) according to the manufacturer's instructions.

## Single-cell RNA sequencing

Fetal lungs were harvested on gestation day 18 from each of two mice that had received water containing DEX or PBS since gestation day 12. Fetal lungs (DEX-1, $n = 8$; DEX-2, $n = 9$; PBS-1, $n = 5$; PBS-2, $n = 8$) were pooled per sample. Fetal lungs cells were prepared by single-cell suspension protocol of fetal lungs. After blocking with CD16/CD32 Monoclonal Antibody, the cells were stained with FVD, antibodies for surface antigens, and Totalseq™-C0301, -C0302, -C0307, and -C0308 (Biolegend, 155861, 155863, 155873, and 155875) for 30 min on ice. The tags were assigned as follows: C0301, fetal lung from DEX-treated mother 1; C0302, fetal lung from DEX-treated mother 2; C0307, fetal lung from PBS-treated mother 1; C0308, fetal lung from PBS-treated mother 2. ILC2s (FVD$^-$ CD45$^+$ Lin$^-$ ST2$^+$) were purified by FACSMelody (BD Biosciences). Libraries were prepared according to the protocol of Chromuim Next GEM Single Cell 5' Reagent Kits v2 (Dual Index) (10x Genomics, PN-1000263) using a Chromium controller (10x Genomics). Briefly, GEMs were first generated by Master Mix with barcoded Single Cell VDJ 5' Gel Beads, cells and distribution to Oil to Chromium Next GEM Chip K (10x Genomics, PN-1000286). Barcoded cDNA was then generated from the GEM. This cDNA was amplified by PCR reaction to generate, 5' Gene Expression library and Feature Barcode library (10x Genomics, PN-1000190, -1000256, -1000215, and -1000250). Sequencing was then performed by Novaseq 6000 (Illumina) at the Laboratory for Research Support of the Institute of Bioregulatory Medicine, Kyushu University.

From the NGS read data, FASTQ files integrated with the Totalseq Hashtags were generated by Cell Ranger Multi function (v7, 10x Genomics). Further analyses were performed using Seurat (version 5.0.1) in the RStudio platform (Version R 4.3.0). From the feature_bc_matrix file from Cell Ranger Multi, Seurat objects were created with the CreateSeuratObject function (min.cells = 3, min.features = 200). Cells with more than 5% of mitochondrial genes and cells with nFeature_RNA less than 200 were filtered out. Expression data were normalized with the NormalizeData function (normalization.method = LogNormalize, scale.factor = 10,000) and scaled with the ScaleData function. Highly variable genes were identified using the FindVariableFeatures function. Principal component analysis (PCA) was performed on the identified highly variable genes using the RunPCA function. Dimensionality reduction was performed using the RunUMAP function, and clustering was performed using the FindClusters function. Differentially expressed genes (DEGs) in each UMAP cluster were identified by the FindAllMarkers function (only.pos = TRUE, min.pct = 0.25, logfc.threshold = 0.25). *Gata3*$^+$ ILC2 was used further analysis.

## Single-nucleus RNA and ATAC sequencing

Fetal lungs were harvested on gestation day 18 from OVA- or PBS-treated mothers and adult lungs were 8 weeks old offspring of OVA- or PBS- treated mothers. (E18-OVA, $n = 8$; E18-PBS, $n = 9$; Adult-OVA-PBS, $n = 6$; Adult-PBS-PBS, $n = 7$). Lung cells were prepared by single-cell suspension protocol of lung tissues. After blocking with CD16/CD32 Monoclonal Antibody, the cells were stained with FVD, antibodies for surface antigens for 15 min on ice. In fetal lung cell collection, ILC2 (CD45$^+$, Lin$^-$, ST2$^+$), immune cells (CD45$^+$) and epithelial cells (EpCAM$^+$) were purified using FACSMelody (BD Biosciences). And in the adult lung cell collection ILC2 (CD45$^+$, Lin$^-$, ST2$^+$), immune cells (CD45$^+$), epithelial cells (EpCAM$^+$), and fibroblasts (EpCAM$^-$, CD31$^-$, PDGFRβ$^+$) were purified by FACSMelody.

The nuclei were isolated according to the protocol of "Nuclei Isolation from Embroynic Mouse Brain for Single Cell Multiome ATAC + Gene Expression Sequencing, Protocol 1" (10x Genomics, CG000366). Briefly, cells were centrifuged at 500 × $g$ for 5 min at 4 °C and removed the supernatant. The pellets were added 100 µl chilled Lysis Buffer (1 mM Tris-HCl (pH 7.4), 1 mM NaCl, 0.3 mM MgCl2, 0.01% Tween-20, 0.01% Nonidet P40 Substitute, 0.001% Digitonin, 0.1% BSA, 0.1 mM DTT, 0.1 U/µl RNase inhibitor), and incubated for 5 min on ice. The lysed cells were washed by 1 ml chilled Wash Buffer three times.

Libraries were prepared according to the protocol of Chromuim Next GEM Single Cell Multiome ATAC + Gene Expression Kit using a Chromium controller (10x Genomics). Briefly, GEMs were first generated by Master Mix with barcoded Single Cell Multiome Gel Beads (10x Genomics, PN-1000285), neuclei and distribution to Oil to Chromium Next GEM Chip J (PN-1000230). Barcoded cDNA was then generated from the GEM. This cDNA was amplified by PCR reaction to generate Gene Expression library and ATAC library (10x Genomics, PN-1000212, PN-1000215). Sequencing was then performed by Novaseq 6000 (Illumina) at the Laboratory for Research Support of the Institute of Bioregulatory Medicine, Kyushu University. From the NGS read data, FASTQ files were generated by CellRanger Arc mkfastq, and then integrated with ATAC and Gene Expression the were generated by Cell Ranger Arc (v2.0.1, 10x Genomics).

Further analysis was performed using Seurat (Version 5.0.1) and Signac (1.12.0) on the RStudio platform (Version R 4.3.0). Seurat objects for gene expression were created from the filtered_feature_bc_matrix file in Cell Ranger Arc with the CreateSeuratObject function (min.cells = 3, min.features = 200). Cells with more than 60% mitochondrial genes and cells with nFeature_RNA less than 200 were filtered out. Expression data were normalized with the NormalizeData function (normalization.method = LogNormalize, scale.factor = 10,000) and scaled with the ScaleData function. In addition, normalization with the SCTransform function was also performed. Highly variable genes were identified using the FindVariableFeatures function; PCA was performed on the identified highly variable genes using the RunPCA function. Dimensionality reduction was performed using the RunUMAP function, and clustering was performed using the FindClusters function. DEGs in each UMAP cluster were identified by the FindAllMarkers function (only.pos = TRUE, min.pct = 0.25, logfc.threshold = 0.25). The cell type was labeled based on the DEGs.

ATAC assays were then created from the filtered_feature_bc_matrix file with the CreateChromatinAssay function. The gene annotation was specified as EnsDb.Mmusculus.v79 with the GetGRangesFromEnsDb function. Quality control was performed with the NucleosomeSignal and TSSEnrichment functions. Filtered with nCount_ATAC < 100,000, nCount_RNA < 25,000, nCount_ATAC > 1000, nCount_RNA > 1000, nucleosome_signal < 2, TSS.enrichment > 1. We called peaks using MACS2 with the CallPeaks function, and processed the DNA accessibility assay with FindTopFeatures function, RunTFIDF function and RunSVD function. The FindTransferAnchors, TransferData, and AddMetaData functions were used to transfer cell type labels from the reference to the query. Both assays were used to create joint neighbor graphs; dimension reduction was performed using the RunUMAP function. ILC2, Type II alveolar cell (AT2), and Macrophages were used for further analysis. For the identified peak genomic regions, sequence motif information was assessed in the UCSC Genome Browser on Mouse (GRCm38/mm10) using the JASPAR 2024 Transcription Factor Binding Site Database.

## bulkATAC sequencing

Fetal lungs were harvested on gestation day 18 from *Alternaria*- or PBS-treated mothers. (E18-*Alternaria*, $n = 8$, from 1 dam; E18-PBS, $n = 18$, from 2 dams). Lung cells were prepared by single-cell suspension protocol of lung tissues. After blocking with CD16/CD32 Monoclonal Antibody, the cells were stained with FVD, antibodies for surface antigens for 15 min on ice. In fetal lung cell collection, ILC2 (CD45$^+$, Lin$^-$, ST2$^+$, CD90.2$^+$) were purified using FACSMelody (BD Biosciences).

bulkATAC-Seq was performed using 2,000 cells/sample. Briefly, Cells were processed using an ATAC-Seq Kit (Active Motif, 53150) following the manufacturer's protocol. The library was purified, and then samples underwent quality control and quantification on an Agilent

2100 Bioanalyzer (Agilent) and sequencing was then performed by Novaseq 6000 (Illumina) at the Laboratory for Research Support of the Institute of Bioregulatory Medicine, Kyushu University.

All sequencing reads after trimming by TrimGalore (version 0.6.10) and Cutadapt (version 4.4) were aligned to the mouse and genome sequences (mouse genome; Genome Reference Consortium Mouse Build 38, mm10, Ensembl v.98) using Bowtie 2 (version 2.5.2). SAMtools (version 1.18) was used to manipulate alignments in the SAM and BAM formats. For visualization of ATAC-seq using the Integrative Genomics Viewer (version 2.17.0), genome coverage tracks were generated using bamCoverage from deepTools (version 3.5.4) with the parameters "--binSize 10 --normalizeUsing RPGC".

### H3K4me3 Carrier Assisted ChIP-seq (CATCH-seq)

CATCH-seq is an improved method of the ultra low-input native ChIP-seq (ULI-NChIP)[68], and the original protocol of CATCH-seq was described in previous reports[69]. We have made some changes in CATCH-seq to improve the usability.

Snap frozen cell pellets from ~3000 ILC2 with 2–3 μL carryover were lysed by 17 μL of cold Nuclei EZ lysis buffer (Sigma, NUC-101) supplemented with complete EDTA-free protease inhibitor cocktail and 1 mM phenylmethanesulfonyl fluoride. Then, 1 μL of 1% Triton X-100 (Merck 93443) and 1% deoxycholate (Nacalai, 10712-54) mixture solution was added to the samples, and sit on ice for 5 min. The chromatin was fragmented by 4 U/μl MNase (NEB, M0247S) in 1xMNase buffer supplemented with 1% PEG6000 (HAMPTON RESEARCH, HR2-533) and 2 mM DTT (Nacalai) at 37 C for 7.5 min. The MNase reaction was stopped by adding 1/10 volume of 100 mM EDTA and 1/12 volume of the 1% Triton X-100 and 1% deoxycholate mixture, and the samples were rest on ice for 15 min. The chromatin lysates were then added with freshly prepared immunoprecipitation buffer[68], and 5% volume was kept for input library construction. To the rest of the lysate, 30 ng of annealed I-SceI carrier DNA was added[67,69]. The forward and reverse strands of the carrier DNA were as follows: /5AmMC6/Gtagggataa-cagggtaattagggataacagggtaattagggataacagggtaattagggataacagggtaatt agggataacagggtaattagggat aacagggtaat\*c/3AmMO/ and /5AmMC6/Gattaccctgttatccctaattaccctgttatccctaattaccctgttatccctaattaccctgt-tatccctaattaccctgttatccctaattaccctgttatccctaattaccctgttatccc cta\*c/3AmMO/, respectively, where asterisks represent phosphorothioate bonds. The oligos were synthesized by Integrated DNA Technologies. For each immunoprecipitation reaction, 0.25 μL of rabbit anti-H3K4me3 (Active Motif 39159) antibody conjugated to precleared Dynabeads Protein A (Thermo Fisher Scientific, 10006D) and G (Thermo Fisher Scientific, 10007D) mixture was used. After immunoprecipitation at 4 C overnight, the chromatin-Dynabeads were washed twice each by the low and high salt wash buffers, and the chromatin was eluted in the freshly prepared ChIP elution buffer at 65 C for 1 h[68]. DNA was recovered by phenol-chloroform extraction followed by ethanol precipitation. Adapter ligation was performed by NEBNext Ultra II DNA Library Prep Kit for Illumina (E7645, NEB) in a half scale of the manufacturer's instruction, and the libraries were purified by 1.8x SPRIselect beads (B23318, Beckman Coulter). The DNA was amplified by KAPA Hifi 2X mater PCR mix (KK2605) for 16 PCR cycles with dual indexing primers (NEBNext Multiplex Oligos for Illumina, E6440). After purification with 0.9x SPRIselect beads, the samples were digested by I-SceI (5 U/μl, NEB, R0694) 37 C for 2 h followed by heat inactivation at 65 C for 20 min and purified by 0.9x SPRIselect beads. The second amplification was not performed. The libraries were sequenced on Novaseq 6000 (Illumina) with paired-end reads.

### H3K4me3 ChIP-seq data analyses

All sequencing reads after trimming by TrimGalore (version 0.6.10) and Cutadapt (version 4.4) were aligned to the mouse and genome sequences (mouse genome; Genome Reference Consortium Mouse Build 38, mm10, Ensembl v.98) using Bowtie 2 (version 2.5.2) with the

"-q -N 1 -L 25 --no-mixed --no-discordant" options. After the removal of PCR duplicates using Sambamba (version 1.0.0) with the parameters "markdup -r -t 2". SAMtools (version 1.18) was used to manipulate alignments in the SAM and BAM formats. For visualization of ChIP-seq using the Integrative Genomics Viewer (version 2.17.0), genome coverage tracks were generated using bamCoverage from deepTools (version 3.5.4) with the parameters "--scaleFactor Scaling_factor --binSize 50 --normalizeUsing RPKM".

### Statistics

Details of statistical analyses and methods for each experiment are provided in the figure legends. Statistical analyses were performed using GraphPad Prism (Version 9). Data distribution was assumed to be normal, but this was not formally tested. $p < 0.05$ was considered statistically significant.

### Reporting summary

Further information on research design is available in the Nature Portfolio Reporting Summary linked to this article.

## Data availability

Single cell RNA-seq data for feral lung ILC2 from DEX- or PBS-treated mothers (GSE262718), single nucleus RNA-seq (GSE262715), ATAC-seq (GSE262716) and ChIP-seq (GSE262949) data for lung cells from OVA- or PBS-treated mothers are available. All other data are available in the article and its Supplementary files or from the corresponding author upon request. Source data are provided with this paper.

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

## Acknowledgements

We thank the members of the Ito lab for their technical assistance and helpful discussions. We thank Kazuyo Moro and Takuya Yashiro for their technical assistance and helpful discussions. We thank Ichiro Taniuchi, Akihiko Yoshimura and Kazufumi Kunimura for the helpful discussions. We thank Hans-Reimer Rodowald for providing *Il7r*-Cre mice. We thank Koichi Ikuta for providing GR-flox mice. We thank Laboratory for Research Support, Medical Institute of Bioregulation at Kyushu University for maintenance of various research equipment for our use and NGS analysis. This work was supported in part by the MEXT Cooperative Research Project Program, Medical Research Center Initiative for High Depth Omics, and CURE:JPMXP1323015486 for MIB, Kyushu University. This work was funded by JSPS KAKENHI 21H02719, 21H00432, 21H05044, 21K19382, 22H05061 and 23H04785, MOON SHOT 21zf0127003, AMED-PRIME 22gm6210012 and JP20gm6110012 (to A.I.), AMED- 22wm0425011, JST-21470411, Research grant from the Chemo-Sero-Therapeutic Research Institute, the Kishimoto Family Foundation, the Mitsubishi Foundation, the Uehara Memorial Foundation, the Inamori Foundation, the Foundation of Kinoshita Memorial Enterprise, Kishimoto Family Foundation, and the NOVARTIS Foundation (Japan) for the Promotion of Science (M.I.).

## Author contributions

T.T. and M.I. conceived the study and designed the experiments. T.T., M.I., and A.M. performed the experiments and analyzed the data. T.T., M.I., C.K., and M.S. analyzed the scRNA-seq data, the snRNA-seq data, and the ATAC-seq data. A.I. performed CATCH-seq. T.T., K.K., I.O., and M.I. wrote the manuscript. All the authors edited the manuscript.

## Competing interests

The authors declare no competing interests.
