## [Transparent Peer Review file · Nature Communications]

Maternal asthma imprints fetal lung ILC2s via glucocorticoid signaling leading to worsened allergic airway inflammation in murine adult offspring

Corresponding Author: Dr Minako Ito

Version 0:

Reviewer comments:

Reviewer #1

(Remarks to the Author)

This study examined the effect of maternal allergen exposure on type 2 responses in the fetal and postnatal lung. Female mice were sensitized to PBS/Alum or OVA/Alum and then challenged with OVA while pregnant. Offspring were challenged with PBS or non-specific HDM allergen. OVA-HDM mice had a greater number of eosinophils and ILC2s that produced IL-5 and IL-13 at greater levels in response to IL-33 than controls. Single nuclei RNA seq analysis identified significant gene expression changes in the ILC2 cluster of OVA fetal lungs, including the glucocorticoid receptor (GR), GATA3, and IL7R. Interestingly, increased GR expression in OVA dams was unique to fetal lung ILC2s. Single cell ATAC-seq identified increased chromatin accessibility in the GATA3 gene that included a GR motif binding site. Maternal dexamethasone alone substantially increased eosinophil and ILC2 levels in offspring, which was augmented by HDM challenge. These findings suggest corticosteroids contribute to expansion of ILC2 populations in maternal allergen exposures with further persistence in offspring. The findings are intriguing with little known about how maternal asthma contributes to development of asthma in young children. It also raises questions about how antenatal steroids to pregnant mothers may alter fetal and infant immunity, particularly type 2-skewing. However, the role for GR and MAPK/AP-1 is less convincing since it is unclear whether the open chromatin region at GATA3 gene has any functional role in ILC2s.

Major Comments

1. Increased GR gene expression and enrichment for MAPK signaling in sequencing data does not necessarily indicate GR and AP-1 binding to GATA3 gene and regulating its expression. More evidence would be needed to confirm that increased GATA3 expression/activity in ILC2 is regulated via this mechanism. Thus, related conclusions (in lines 244-249) are speculative.
2. Of note, GATA3 tracks in Fig 5F appears to have additional regions with greater accessibility in allergen mice. It is interesting that this appears to persist in OVA adult mice. Are regions changed in further upstream the GATA3 promoter?
3. The effect of corticosteroids appear to be specific for ILC2s but would they regulate GATA3 in Th2 cells similarly? Are chromatin regions in GATA3 gene closed in Th2 cells?
4. How does ILC2 respond to corticosteroids ex vivo? Is IL-33-induced IL-5 or IL-13 resistant to corticosteroids?
5. The level of allergic airway inflammation in the offspring lung is not evident with no histology data provided. Is ILC2 activation in the OVA-HDM sufficient to induce peri-bronchiolar and/or -vascular inflammation?
6. In lines 71-74, this sentence could be stated more clearly.
7. In lines 131-133, the "trend toward increased numbers" should be modified to accurately reflect the data. The data either significantly increased or not significantly changed.
8. In lines 208-210, what is meant by "phenotypic changes"?

Reviewer #2

(Remarks to the Author)

In this manuscript, the authors aimed to investigate intergenerational non-antigen-specific asthma susceptibility transfer in

mice. By utilizing different allergens given to the mothers and offspring, the authors found that OVA-systemic sensitized and challenged asthmatic female mice transferred persistently enhanced ILC2 activation, eosinophilic lung inflammation, and airway hyperresponsiveness in offspring mice challenged with house dust mite (HDM). Offspring ILC2 CHIPseq analysis identified increased chromatin accessibility to GATA3 and single-cell RNA-seq analysis of offspring ILC2s showed that Nr3c1 (encodes glucocorticoid receptor) was increased in ILC2s from OVA-sensitized mothers. Further, OVA-sensitized mothers had elevated corticosteroids (CS) and dexamethasone-exposed mothers transferred a similar propensity for heightened ILC2 activation in offspring challenged with HDM to that of OVA-sensitized mothers. The manuscript is well written with potentially impactful findings related to how maternal CS or antigen-driven type 2 inflammation leads to enhanced offspring ILC2 responses and lung inflammation to a distinct allergen.

Major

1. Maternal systemic OVA/alum sensitization. Though the OVA/alum model using systemic priming is a useful tool, the human mucosal response to airborne allergens in asthma is likely quite different. A key question is whether the findings related to ILC2 GATA3 accessibility and increased GR are specific to the model where mothers are systemically primed with OVA/alum. Systemic alum alone is well known to induce specific immune responses and doesn't recapitulate the mucosal auto-adjuvant properties of allergens such as fungal allergens or HDM. Optimally, the authors should show whether the model is specific to OVA/alum priming or occurs with mucosal sensitization in mothers.
2. T cell dependence. Though the work shows that CD4 and CD8 cells were depleted in IL-7RKO mice before ILC2 transfer, T cells could still be critical to the outcome (at the time of maternal sensitization), especially in the T cell-dependent OVA model. Based on this, one question that arises is whether the same results would be observed in a RAGKO that has ILC2s but lacks B and T cells? (e.g. Do RAGKO mothers challenged with a fungal allergen such as *Alternaria* followed by offspring challenged with HDM have the same persistently enhanced ILC2 changes? – this would address major points 1 and 2).
3. Combination effects of maternal allergen/DEX. Since DEX and OVA-sensitization had overlapping effects on offspring ILC2s, does treating maternal asthmatic inflammation with DEX lead to further enhancement of T2 inflammation and ILC2 responses in offspring? This is a clinically relevant question given that severe asthma exacerbations are treated with CS in pregnant women.

Minor

1. The term "asthma exacerbation" is used for human asthma and terms for mouse model features should be more precise (e.g. allergen-induced lung inflammation, etc).
2. Figure labels are somewhat confusing. Figure 2a contains both mouse model diagram and results, whereas subsequent panels have these labeled separately (as 2b and 2c). Please be consistent with figure labeling.
3. Consider providing a table that contains the gene sets for activation, proliferation, glucocorticoid signaling, to complement Figure 6g to better understand the overlap analysis.
4. Consider replacing reference 1 with a more updated review of global asthma morbidity.

Reviewer #3

(Remarks to the Author)

In the current report, Takao et al explore the impact of maternal allergen exposure during pregnancy on the development of asthma in offspring. The authors describe a model of maternal allergen exposure wherein mothers are treated intraperitoneally with alum, or OVA + alum 3 weeks, and 1 week prior to mating. 7 – 9 week old adult offspring are given intranasal house dust mite (HDM) and allergic responses assessed. The authors demonstrate a slight increase in lung eosinophils in offspring of allergic mothers in the absence of adult allergen exposure. Offspring of allergic mothers given HDM demonstrated and increase in both lung eosinophil numbers, ILC2 numbers, and airway hyperresponsiveness. ILC2s from adult offspring of allergen exposed moms demonstrated increased capacity to produce Th2 cytokines, and transfer of ILC2s from allergen exposed moms, increased pulmonary eosinophilia. Increased numbers of ILC2s and eosinophils were also observed in fetal lungs in offspring of allergen-exposed moms, and blocking ILC2 recruitment to the fetal lungs (with maternal IL-7R) reversed the increased eosinophil and ILC2 recruitment in the adult lungs. scRNA-seq analysis of ILC2s, immune cells and epithelial cells from fetal lungs revealed pronounced changes in ILC2 transcriptional responses, with increased levels of genes associated with ILC2 activation, MAPK signaling, and glucocorticoid signaling. scATAC-Seq analysis in fetal and adult ILC2 revealed consistent changes in DNA methylation patterns at genes associated with MAPK signaling pathways, and areas demonstrating altered accessibility contain Glucocorticoid responsive elements. Interestingly, the authors also observed increased corticosterone levels on amniotic fluid and fetal serum in offspring of mothers exposed to allergens. Based on this the authors sought to determine whether Glucocorticoid exposure could replicate the impact of maternal allergen exposures on offspring ILC2s numbers. The authors demonstrate that steroid treatment of mothers similarly increased ILC2 inflammatory gene expression, and offspring of steroid treated moms displayed increased capacity for allergen-induced eosinophil, and ILC2 activity/recruitment.

Major Concerns:

As the adult offspring only receive 1 "episode" of HDM exposure, this likely reflects innate-like, early asthma responses which are highly ILC2-dependent. This is reflected by the observations of a relatively minor (i.e. not significant) increases in pulmonary Th2 cells after HDM exposure (compare PBS-PBS and PBS-HDM groups in Fig 1b). Does similar augmentation of eosinophilia and AHR occur in a more "conventional" asthma model where distinct sensitization and challenge phases occur? This could ideally be done if adult HDM exposed animals were rechallenged with HDM 2 – 3 weeks following initial 5 day course of HDM. This is important, as the majority of deleterious asthma outcomes occur after establishment of robust Th2-driven immunity. Early increases in ILC2 number/function may or may not be sufficient to drive such an enhanced Th2-

associated phenotype.

It is unclear whether the authors are really investigating an baseline alteration in the lungs of adult (and fetal) offspring of allergen exposed mothers, or a difference in response to allergen. Given that eosinophils are increased in the adult lungs of OVA offspring, even if not exposed to HDM, and that the AHR data demonstrates a marked shift in AHR between PBS-PBS and OVA-PBS mice, I suspect it may be the former. Can the authors formally test if the HDM-induced change is actually comparable between the control and offspring of allergen-exposed mothers (perhaps by comparing the overall “fold change” in things like eosinophilia, AHR, PC150 in response to HDM)? Given that the authors observe increased fetal eosinophils, and prior reports of fetal eosinophilia and increased baseline AHR (references 19 and 20 cited by the authors), this is an important possibility to be considered.

In Fig 2 the authors show increased responsiveness of sorted ILC2 to IL-33. Do ILC2s show similarly increased responsiveness to IL-25/TSLP? Also – given their later observation of altered MAPK signaling pathway related gene expression, it would be interesting to look at activation of downstream signaling molecules in response to IL-33/IL-25/TSLP (i.e. MAPK, STAT). Is this also enhanced in similarly treated ILC2 from offspring of allergen-exposed moms?

What are the levels of IL-33/IL-25/TSLP in the lungs of fetal and/or adult offspring of allergen exposed moms? Do any of these genes come up in the scRNA-Seq analysis of pulmonary immune and epithelial cell data?

Fig 2e shows increased eosinophil recruitment of IL-33-treated mice given ILC2s from control or offspring of allergen-exposed moms. However, while the authors report that 5K of the ILC2s were given to recipients, it appears that >90% of the transferred cells are not making it to the lung (Supplementary Fig 1 shows only 200 – 400 ILCs in these lungs). Where do the majority of these cells go? Are the authors really comfortable arguing that there is no difference in cell numbers/recruitment given that they are only able to capture the recruitment of <10% of the input cells?

I'm confused about the data shown in Supplementary Figure 2a and 2b. The authors show that numbers of ILC2s in the lung do not differ in fetal mice given IL-7R, but that there are major differences in the numbers of ILC2s present in the lungs of mice 1 week after birth. If this is actually the case (i.e. I am not misunderstanding the data) that would strongly suggest that the typical expansion of ILC2 numbers after birth (or altered recruitment of a later “wave” of ILC2s to the lung) is really what is being impacted. How does this affect the overall interpretation of the authors' data?

The authors demonstrate that the numbers of genes with altered expression in comparing populations of ILC2s from control and offspring of allergen-exposed moms is the largest, but there are significant differences in gene expression in other cell types (macrophages, epithelial cells). What are these differentially expressed genes? Is there any overlap in differentially expressed genes in ILC2s and these other populations?

The observation that corticosterone treatment of mothers can replicate the effect of allergen exposure is interesting. Given that many asthmatic mothers are ALSO on steroids as a treatment for their allergic asthma, this may have profound influences on our understanding of transmission of allergic disease. Is application of both allergen AND steroids synergistic in terms of capacity to augment offspring asthma outcomes? The manuscript would be strengthened by performing such experiments. That the authors did not even mention this aspect of their study is surprising.

Minor concerns:

The graphs in Fig 1b suggest that total CD45+ cells number $\sim 2 \times 10^6$, whereas B cells are close to 1×10^6 . This seems to be an awfully high number of B cells in the lung ($\sim 50\%$ of CD45+ cells). Is this a typo?

Please also show airway responses in the absolute (i.e. not normalized to baseline). It would be helpful to see the actual airway mechanic curves between the groups to allow the reader to better ascertain whether or not the effects of maternal allergen exposure are a result of altered baselines, or altered responsiveness to HDM.

Reviewer #4

(Remarks to the Author)

Version 1:

Reviewer comments:

Reviewer #1

(Remarks to the Author)

I have no further comments.

Reviewer #2

(Remarks to the Author)

The Authors overall addressed my comments. It would be worth discussing one of the points in the revised data, however. The authors added a supplementary figure (Figure S7) illustrating no additional effects of DEX treatment in OVA-challenged mice. Since it is a clinically relevant question (maternal asthma treated by CS), I suggest discussing it further in the discussion section.

Reviewer #3

(Remarks to the Author)

The authors have addressed this reviewer's concerns

Reviewer #4

(Remarks to the Author)

REVIEWER COMMENTS

Reviewer #1 (Remarks to the Author):

This study examined the effect of maternal allergen exposure on type 2 responses in the fetal and postnatal lung. Female mice were sensitized to PBS/Alum or OVA/Alum and then challenged with OVA while pregnant. Offspring were challenged with PBS or non-specific HDM allergen. OVA-HDM mice had a greater number of eosinophils and ILC2s that produced IL-5 and IL-13 at greater levels in response to IL-33 than controls. Single nuclei RNA seq analysis identified significant gene expression changes in the ILC2 cluster of OVA fetal lungs, including the glucocorticoid receptor (GR), GATA3, and IL7R. Interestingly, increased GR expression in OVA dams was unique to fetal lung ILC2s. Single cell ATAC-seq identified increased chromatin accessibility in the GATA3 gene that included a GR motif binding site. Maternal dexamethasone alone substantially increased eosinophil and ILC2 levels in offspring, which was augmented by HDM challenge. These findings suggest corticosteroids contribute to expansion of ILC2 populations in maternal allergen exposures with further persistence in offspring. The findings are intriguing with little known about how maternal asthma contributes to development of asthma in young children. It also raises questions about how antenatal steroids to pregnant mothers may alter fetal and infant immunity, particularly type 2-skewing. However, the role for GR and MAPK/AP-1 is less convincing since it is unclear whether the open chromatin region at GATA3 gene has any functional role in ILC2s.

Major Comments

1. Increased GR gene expression and enrichment for MAPK signaling in sequencing data does not necessarily indicate GR and AP-1 binding to GATA3 gene and regulating its expression. More evidence would be needed to confirm that increased GATA3 expression/activity in ILC2 is regulated via this mechanism. Thus, related conclusions (in lines 244-249(→272-275)) are speculative.

Thanks for the important comment. To evaluate the function of GR in ILC2s, we analyzed mice deficient in GR in ILC2s (*Cd127^{cre/+} Nr3c1^{fl/fl}*). GATA3 expression was impaired in lung ILC2s of these mice, indicating that GR regulation is important for GATA3 expression. We provided the data in **Fig. 5g**, and added to the text (in lines 272-275). We understand the reviewer's concern, and we modified the text accordingly. Specifically, we have removed the sentence stating, "It has also been reported that the effects of glucocorticoids can persist well beyond the duration of their exposure, suggesting that antenatal

glucocorticoid signaling may be involved in the long-term increased chromatin accessibility of the *Gata3* locus in adult offspring lung ILC2s.” Additionally, we have rephrased our interpretation in lines 247-249 to, “These suggest that maternal asthma can have long-lasting effects on fetal lung ILC2s, leading to epigenetic changes in adult offspring.” This revised statement now appears in lines 275-277. Thanks for highlighting this point, which helped us refine our conclusions.

2. Of note, GATA3 tracks in Fig 5F appears to have additional regions with greater accessibility in allergen mice. It is interesting that this appears to persist in OVA adult mice. Are regions changed in further upstream the GATA3 promoter?

Thanks for this observation regarding the GATA3 tracks in Fig. 5F. As open chromatin regions common to the fetus and adult offspring, two peaks were detected in the *Gata3* gene locus in addition to the GR-binding region (A).

(A): scATAC-seq and ChIP-seq results in lung ILC2s of offspring from asthmatic or control mothers in the *Gata3* locus.

An additional upstream region regulating GATA3 in ILC2s has been previously reported (Furuya H, *et al.* Nat Commun. 2024, PMID: 38969652; Kasal DN, *et al.* Proc Natl Acad Sci U S A. 2021, PMID: 34353913). In our single-cell ATAC-seq analysis, we were unable to detect notable peaks in this region to present as definitive data, but a slight increase in peak intensity was observed in the offspring of asthmatic mothers (B).

B

(B): scATAC-seq results in lung ILC2s of offspring, showing a broader region around the *Gata3* locus, including SE1 and SE2 regions previously reported as regulatory elements for *Gata3*.

Furuya, H., Toda, Y., Iwata, A. et al. Stage-specific GATA3 induction promotes ILC2 development after lineage commitment. *Nat Commun* **15**, 5610 (2024). <https://doi.org/10.1038/s41467-024-49881-y>

Figure adapted from [Furuya H, et al. *Nat Commun*. 2024, PMID: 38969652].

[figure redacted]

Figure adapted from [Kasal DN, et al. *Proc Natl Acad Sci U S A*. 2021, PMID: 34353913].

3. The effect of corticosteroids appear to be specific for ILC2s but would they regulate GATA3 in Th2 cells similarly? Are chromatin regions in GATA3 gene closed in Th2 cells?

Our single-cell ATAC-seq analysis revealed no significant changes in the *Gata3* locus in adult offspring from a maternal OVA asthma model, compared to controls and total T cells. However, the limited number of T cells available for analysis made it difficult to make meaningful comparisons based on Th2 cell identification. Therefore, to investigate the effect of corticosteroids on Th2 cells in the offspring, we analyzed lung T cells from adult offspring of DEX-treated mothers using single-cell ATAC-seq. It was difficult to assess which group had more open chromatin at the *Gata3* locus, as accessibility varied across regions within the locus. However, with regard to the *Nr3c1* binding region, where changes were observed in the lung ILC2 of the offspring from asthmatic mothers, there was also a slight increase in the peak in the lung Th2 of adult offspring from DEX-treated mothers. These findings suggest that corticosteroids may also impact the Th2 cells in the lungs of adult offspring. Considering that CD4⁺ T cells in *Nr3c1*^(-/-) mice have been reported to have a reduced capacity to produce type 2 cytokines (Shimba A, *et al.* Immunity. 2018, PMID: 29396162), it is possible that corticosteroids are involved in functional changes in Th2 cells through epigenetic mechanisms.

(A): Experimental design for scATAC-seq. CD4⁺ cells were sorted from adult offspring from DEX-treated or control mothers. (B): UMAP visualization of scATAC-seq results, highlighting Th2 cells outlined in red. (C): scATAC-seq results in lung Th2 cells of adult offspring from DEX-treated or control mothers in the *Gata3* locus.

4. How does ILC2 respond to corticosteroids ex vivo? Is IL-33-induced IL-5 or IL-13 resistant to corticosteroids?

In previous reports, corticosteroids have been shown to inhibit IL-33-induced cytokine production by ILC2s in vitro (Kabata H, *et al.* Nat Commun. 2013, PMID: 24157859). Consistent with these findings, when we sorted ILC2s from the lungs of untreated wild-type C57BL/6J mice and stimulated them with IL-33 following DEX exposure in culture, cytokine production was reduced. To further investigate the effect of corticosteroid exposure on ILC2s in vivo, we sorted and cultured lung ILC2s from adult offspring of DEX-treated mothers. These ILC2s tended to produce more cytokines than those from the control group. Different effects of corticosteroids on ILC2s have been reported in vivo and in vitro (Feng B, *et al.* Front Immunol. 2022, PMID: 36032134), and our experimental results support this divergence.

(A): In vitro effects of DEX exposure on ILC2s from untreated wild-type C57BL/6J mice. (B): Lung ILC2s were sorted from adult offspring of DEX-treated or control mothers and cultured in the presence of IL-33 for 3 days. IL-5 and IL-13 levels in the culture supernatants of sorted lung ILC2s were measured by ELISA.

5. The level of allergic airway inflammation in the offspring lung is not evident with no histology data provided. Is ILC2 activation in the OVA-HDM sufficient to induced peri-bronchiolar and/or -vascular inflammation?

We assessed the extent of lung tissue inflammation in adult offspring of asthmatic mothers by performing hematoxylin-eosin (H&E) staining of lung sections. The results showed an increased degree of peribronchiolar inflammation in lung tissue from the OVA-HDM group compared to the PBS-HDM group. We provided the data in **Fig. 1e**, and added to the text (in lines 116-118).

6. In lines 71-74, this sentence could be stated more clearly.

Thanks for the reviewer's suggestions. We have revised the sentence in lines 71-74 to improve clarity (in lines 71-74).

7. In lines 131-133 (→145-149), the “trend toward increased numbers” should be modified to accurately reflect the data. The data either significantly increased or not significantly changed.

We have revised the text in lines 131-133 (→145-149) to accurately reflect the data by removing the phrase 'trend toward increased numbers' and clarified that there were no significant changes (in lines 145-149).

8. In lines 208-210 (→223-224), what is meant by “phenotypic changes”?

We modified the text to state: “asthma during pregnancy upregulates genes related to activation and glucocorticoid signaling in fetal lung ILC2s.” (in lines 223-224). This revision eliminates the ambiguity surrounding the term “phenotypic changes”.

Reviewer #2 (Remarks to the Author):

In this manuscript, the authors aimed to investigate intergenerational non-antigen-specific asthma susceptibility transfer in mice. By utilizing different allergens given to the mothers and offspring, the authors found that OVA-systemic sensitized and challenged asthmatic female mice transferred persistently enhanced ILC2 activation, eosinophilic lung inflammation, and airway hyperresponsiveness in offspring mice challenged with house dust mite (HDM). Offspring ILC2 CHIPseq analysis identified increased chromatin accessibility to GATA3 and single-cell RNA-seq analysis of offspring ILC2s showed that *Nr3c1* (encodes glucocorticoid receptor) was increased in ILC2s from OVA-sensitized mothers. Further, OVA-sensitized mothers had elevated corticosteroids (CS) and dexamethasone-exposed mothers transferred a similar propensity for heightened ILC2 activation in offspring challenged with HDM to that of OVA-sensitized mothers. The manuscript is well written with potentially impactful findings related to how maternal CS or antigen-driven type 2 inflammation leads to enhanced offspring ILC2 responses and lung inflammation to a distinct allergen.

Major

1. Maternal systemic OVA/alum sensitization. Though the OVA/alum model using systemic priming is a useful tool, the human mucosal response to airborne allergens in asthma is likely quite different. A key question is whether the findings related to ILC2 GATA3 accessibility and increased GR are specific to the model where mothers are systemically primed with OVA/alum. Systemic alum alone is well known to induce specific immune responses and doesn't recapitulate the mucosal auto-adjuvant properties of allergens such as fungal allergens or HDM. Optimally, the authors should show whether the model is specific to OVA/alum priming or occurs with mucosal sensitization in mothers.

Thanks for the reviewer's comments. We tested the effects on the offspring of a model of asthma caused by mucosal sensitization of pregnant mice through intranasal administration of *Alternaria*. Fetuses of *Alternaria* asthmatic mothers showed increased proportions of eosinophils and ILC2 in the lung compared to control subjects, exhibiting similar changes to those observed in the OVA asthma model. GR levels were significantly elevated in fetal lung ILC2s, similar to the OVA asthma model, and a trend toward higher GATA3 expression was also observed. We further sorted these fetal lung ILC2s for bulk ATAC-seq analysis and found an increased open chromatin region at the *Gata3* gene locus in fetal lung ILC2s from *Alternaria* asthmatic mothers. Additionally, we observed a slight increase in the numbers of lung eosinophils and cytokine-producing ILC2s was observed in adult offspring of *Alternaria* asthmatic

mothers after intranasal administration of house dust mite (HDM). We provided the data in **Supplementary Fig. 6a–d**, and added to the text (in lines 277-292).

These experimental results suggest that similar effects to those observed in the OVA asthma model are present in the *Alternaria* mucosal sensitization asthma model in their offspring. However, the effects observed in adult offspring from the *Alternaria* model were less pronounced compared to the OVA asthma model, which may be due to the more systemic priming nature of the OVA/alum model.

2. T cell dependence. Though the work shows that CD4 and CD8 cells were depleted in IL-7RKO mice before ILC2 transfer, T cells could still be critical to the outcome (at the time of maternal sensitization), especially in the T cell-dependent OVA model. Based on this, one question that arises is whether the same results would be observed in a RAGKO that has ILC2s but lacks B and T cells? (e.g. Do RAGKO mothers challenged with a fungal allergen such as *Alternaria* followed by offspring challenged with HDM have the same persistently enhanced ILC2 changes? – this would address major points 1 and 2).

Thanks for the reviewer's insightful comment regarding the use of RAGKO mice to investigate the T cell dependency of the observed effects. We agree that employing RAGKO mice, which lack both B and T cells, would indeed offer valuable insights into the role of T cells in our findings. However, we were unable to obtain a sufficient number of RAGKO mice to perform experiments that could fully address all aspects of the reviewer's request. Furthermore, considering the critical role of acquired immunity in asthma models induced by allergens such as fungal species or OVA (Corry DB, et al. Mol Med. 1998, PMID: 9642684; Doherty TA, et al. Am J Physiol Lung Cell Mol Physiol. 2009, PMID: 19060225), we believed it would be challenging to induce sufficiently chronic asthma in pregnant RAGKO mice, as was achieved in our study.

As an alternative approach, we conducted experiments to assess T cell dependency in the worsening of asthma in adult offspring from asthmatic mothers.

First, we crossed *Rag2*^(+/+) females with *Rag2*^(-/-) males and induced chronic asthma during pregnancy through HDM sensitization and exposure. Then, we compared the response to IL-33 administration between adult *Rag2*^(-/-) offspring from asthmatic mothers and those from control mothers. In this model, *Rag2*^(+/+) mother mice are able to induce sufficient asthma because they possess both T cells and B cells. In contrast, *Rag2*^(-/-) adult offspring lack T and B cells, allowing us to evaluate whether the worsening of asthma in adult offspring from asthmatic mothers is dependent on acquired immunity in the offspring. The data showed that there was a tendency for the number of eosinophils and the number of cytokine-producing

ILC2s to increase in the lungs of adult children born to mothers who had developed HDM-induced asthma.

Second, CD4 and CD8 cells were depleted through antibody treatment in adult offspring from *Alternaria* asthmatic mothers, and their response to HDM was assessed. This model allows for the evaluation of T cell-independent reactivity to HDM in adult offspring. The data suggested a trend toward an elevation in the number of cytokine-producing ILC2s in the lungs of adult offspring from mothers with asthma induced by *Alternaria*. Although the number of eosinophils in the lungs was not significantly altered, it is possible that eosinophilic inflammation was not sufficiently induced in the antibody-treated mice, considering the crucial role that T cells play in HDM-induced asthma models (Coquet JM, et al. *Immunity*. 2015, PMID: 26287681; Li BW, et al. *Eur J Immunol*. 2016, PMID: 27062360).

In both experimental results, conditions lacking T cells in adult offspring from asthmatic mothers did not show significant changes in lung eosinophil counts or a significant increase in the number of cytokine-producing ILC2s, although there was a trend toward an increase. These findings suggest that changes in T cells may play a role in the worsening of asthma in adult offspring from asthmatic mothers. However, this knowledge does not contradict our study's main finding that maternal asthma increases the responsiveness of lung ILC2s in adult offspring.

(A): Female Rag2^(+/+) mice were bred with male Rag2^(-/-) mice and treated intranasally with HDM or PBS during pregnancy to induce asthma. Adult offspring were intranasally stimulated with 0.5 µg of IL-33 for two days. (B): Eosinophil counts and cytokine-producing ILC2 numbers in the lungs of Rag2^(-/-) adult offspring from asthmatic or control mothers. (C): Model of adult offspring from Alternaria asthmatic or control mothers, in which CD4 and CD8 cells were depleted by antibody treatment. Adult offspring were treated intranasally with HDM. (D): Eosinophil counts and cytokine-producing ILC2 numbers in the lungs of adult offspring from Alternaria asthmatic or control mothers.

3. Combination effects of maternal allergen/DEX. Since DEX and OVA-sensitization had overlapping effects on offspring ILC2s, does treating maternal asthmatic inflammation with DEX lead to further enhancement of T2 inflammation and ILC2 responses in offspring? This is a clinically relevant question given that severe asthma exacerbations are treated with CS in pregnant women.

Thanks for the reviewer's insightful comment regarding the potential interactions between allergen exposure and corticosteroid treatment in asthmatic mothers. We induced chronic asthma in pregnant mice and concurrently administered oral dexamethasone (DEX) from E12 onwards. Our data indicate that there were no significant differences in lung eosinophil numbers between the adult offspring of mothers in the DEX combination group and those in the allergen-only group. While there were no major changes in cytokine-producing ILC2s, there was a tendency for an increased number of ILC2s in the adult offspring of mothers receiving DEX. Additionally, in another experiment using non-pregnant mice to assess the effects of DEX on our OVA/Alum asthma model, we found no significant differences in eosinophilic inflammation in the lungs between the two groups. These results suggest that, within the context of our experimental model, DEX treatment did not alleviate lung inflammation in the mothers, which likely reflects the lack of changes in eosinophil numbers in their adult offspring. However, we observed a trend towards an increase in lung ILC2 numbers in the offspring of mothers treated with DEX, indicating that systemic administration of steroids to asthmatic mothers may synergistically influence lung ILC2s in their adult offspring alongside the effects of asthma itself. These results have been added to **Supplementary Fig. 7c–e**, and corresponding text has been included in the manuscript (in lines 329-347).

Minor

1. The term “asthma exacerbation” is used for human asthma and terms for mouse model features should be more precise (e.g. allergen-induced lung inflammation, etc).

Thanks for the reviewer's feedback. We have revised the manuscript to replace the term "asthma exacerbation" with more precise terminology, primarily using "allergen-induced lung inflammation" to describe the features observed in the mouse models.

2. Figure labels are somewhat confusing. Figure 2a contains both mouse model diagram and results, whereas subsequent panels have these labeled separately (as 2b and 2c). Please be consistent with figure labeling.

Thanks for the reviewer's comment. We have combined Fig. 2b and 2c into Fig. 2b, and Fig. 2d and 2e into Fig. 2c to ensure consistent labeling throughout the figure. Additionally, we have corrected the corresponding sections in the text.

3. Consider providing a table that contains the gene sets for activation, proliferation, glucocorticoid signaling, to complement Figure 6g to better understand the overlap analysis.

We summarized the gene sets used in the analyses presented in Fig. 6g and Supplementary Fig. 4d, and provided them in **Supplementary Table 3**.

4. Consider replacing reference 1 with a more updated review of global asthma morbidity.

Thanks for the reviewer's suggestion. We replaced reference 1 with a more updated source and revised the sentence in the manuscript to reflect the most recent data.

Reviewer #3 (Remarks to the Author):

In the current report, Takao et al explore the impact of maternal allergen exposure during pregnancy on the development of asthma in offspring. The authors describe a model of maternal allergen exposure wherein mothers are treated intraperitoneally with alum, or OVA + alum 3 weeks, and 1 week prior to mating. 7 – 9 week old adult offspring are given intranasal house dust mite (HDM) and allergic responses assessed. The authors demonstrate a slight increase in lung eosinophils in offspring of allergic mothers in the absence of adult allergen exposure. Offspring of allergic mothers given HDM demonstrated an increase in both lung eosinophil numbers, ILC2 numbers, and airway hyperresponsiveness. ILC2s from adult offspring of allergen exposed moms demonstrated increased capacity to produce Th2 cytokines, and transfer of ILC2s from allergen exposed moms, increased pulmonary eosinophilia. Increased numbers of ILC2s and eosinophils were also observed in fetal lungs in offspring of allergen-exposed moms, and blocking ILC2 recruitment to the fetal lungs (with maternal α IL-7R α) reversed the increased eosinophil and ILC2 recruitment in the adult lungs. scRNA-seq analysis of ILC2s, immune cells and epithelial cells from fetal lungs revealed pronounced changes in ILC2 transcriptional responses, with increased levels of genes associated with ILC2 activation, MAPK signaling, and glucocorticoid signaling. scATAC-Seq analysis in fetal and adult ILC2 revealed consistent changes in DNA methylation patterns at genes associated with MAPK signaling pathways, and areas demonstrating altered accessibility contain Glucocorticoid responsive elements. Interestingly, the authors also observed increased corticosterone levels on amniotic fluid and fetal serum in offspring of mothers exposed to allergens. Based on this the authors sought to determine whether Glucocorticoid exposure could replicate the impact of maternal allergen exposures on offspring ILC2s numbers. The authors demonstrate that steroid treatment of mothers similarly increased ILC2 inflammatory gene expression, and offspring of steroid treated moms displayed increased capacity for allergen-induced eosinophil, and ILC2 activity/recruitment.

Major Concerns:

As the adult offspring only receive 1 “episode” of HDM exposure, this likely reflects innate-like, early asthma responses which are highly ILC2-dependent. This is reflected by the observations of a relatively minor (i.e. not significant) increases in pulmonary Th2 cells after HDM exposure (compare PBS-PBS and PBS-HDM groups in Fig 1b). Does similar augmentation of eosinophilia and AHR occur in a more “conventional” asthma model where distinct sensitization and challenge phases occur? This could ideally be done if adult HDM exposed animals were rechallenged with HDM 2 – 3 weeks following initial 5 day

course of HDM. This is important, as the majority of deleterious asthma outcomes occur after establishment of robust Th2-driven immunity. Early increases in ILC2 number/function may or may not be sufficient to drive such an enhanced Th2-associated phenotype.

In our study, we adopted a deliberately suboptimal protocol for allergic exposure in adult offspring to observe the effects of maternal asthma, drawing on the strategies from previous reports (Hamada K, *et al.* J Immunol. 2003, PMID: 12574331). This approach aimed to prevent saturating the response with excessive allergy-induced lung inflammation in adult offspring, which could obscure differences between groups. While we sincerely appreciate the reviewer's important points, and in consideration of the context discussed earlier, we conducted additional experiments using a model of HDM-induced allergic lung inflammation, with sensitization on day 0 and exposure on days 7-11. Previous reports indicate that Th2 cells are activated with this protocol (Li BW, *et al.* Eur J Immunol. 2016, PMID: 27062360). The data revealed a trend toward increased eosinophil numbers in the lungs of adult offspring from asthmatic mothers. The number of ILC2s in the lungs was significantly elevated, along with the population of cytokine-producing ILC2s. In contrast, the number of Th2 cells and cytokine-producing Th2 cells in the lungs exhibited an increasing trend but did not show significant changes. These results suggest that even with the sensitization and exposure protocol to HDM, ILC2s may be robustly activated in the lungs of adult offspring from asthmatic mothers. Th2 changes were not significant, but there was a trend towards an increase, and given previous reports on the co-relationship between ILC2 and Th2 (Martinez-Gonzalez I, *et al.* Trends Immunol. 2015, PMID: 25704560), it is possible that ILC2 changes are involved in the enhancement of the Th2-related phenotype, as the reviewer noted. We included these results in **Supplementary Fig. 2a** and added to the text (in lines 128-137).

It is unclear whether the authors are really investigating an baseline alteration in the lungs of adult (and fetal) offspring of allergen exposed mothers, or a difference in response to allergen. Given that eosinophils are increased in the adult lungs of OVA offspring, even if not exposed to HDM, and that the AHR data demonstrates a marked shift in AHR between PBS-PBS and OVA-PBS mice, I suspect it may be the former. Can the authors formally test if the HDM-induced change is actually comparable between the control and offspring of allergen-exposed mothers (perhaps by comparing the overall “fold change” in things like eosinophilia, AHR, PC150 in response to HDM)? Given that the authors observe increased fetal eosinophils, and prior reports of fetal eosinophilia and increased baseline AHR (references 19 and 20 cited by the authors), this is an important possibility to be considered.

Thanks for the reviewer's thoughtful comment. We appreciate your insights into the baseline changes in the lungs of offspring of allergen-exposed mothers. To address your suggestion, we calculated the ratio of changes between the groups as follows: (OVA-HDM/OVA-PBS)/(PBS-HDM/PBS-PBS). Importantly, the results showed that the 95% confidence intervals for these ratios crossed 1, indicating that the HDM-induced changes are not significantly different between the control and the offspring of allergen-exposed mothers, thereby addressing your concern.

The ratio of airway hyperresponsiveness (AHR), PC150 and eosinophilia, calculated as (OVA-HDM/OVA-PBS)/(PBS-HDM/PBS-PBS), with 95% confidence intervals indicated.

In Fig 2 the authors show increased responsiveness of sorted ILC2 to IL-33. Do ILC2s show similarly increased responsiveness to IL-25/TSLP? Also – given their later observation of altered MAPK signaling pathway related gene expression, it would be interesting to look at activation of downstream signaling molecules in response to IL-33/IL-25/TSLP (i.e. MAPK, STAT). Is this also enhanced in similarly treated ILC2 from offspring of allergen-exposed moms?

We performed in vitro experiments where sorted ILC2s were stimulated with IL-25, IL-25 + IL-2, TSLP + IL-2 and TSLP + IL-25. However, we found that IL-5 and IL-13 levels were below the detection limit in all conditions, and there were no differences between lung ILC2s from adult offspring of allergen-exposed mothers and controls (data not shown). Regarding the downstream signaling molecules activated by IL-33, we investigated the activation of the MAPK pathway. In adult offspring of allergen-exposed mothers, we observed increased phosphorylation of p38 in lung ILC2s at baseline. Furthermore, after in vivo stimulation with IL-33, the phosphorylation of p38 was significantly increased compared to controls. We included these results in **Supplementary Fig. 5e** and added to the text (in lines 262-266).

What are the levels of IL-33/IL-25/TSLP in the lungs of fetal and/or adult offspring of allergen exposed moms? Do any of these genes come up in the scRNA-Seq analysis of pulmonary immune and epithelial cell data?

We evaluated the cytokine levels of IL-33, IL-25 and TSLP in both fetal and adult offspring. Regarding fetal lungs, we were only able to measure IL-33, while IL-25 and TSLP levels were below the detection limit, with no significant differences observed between the offspring of allergen-exposed mothers and controls. For IL-33, the experimental results were not reproducible, making it difficult to determine a clear difference between the fetal lungs of offspring from asthmatic mothers and those from the control group. In adult offspring, we found that IL-33 levels were significantly elevated during HDM exposure. This suggests that, in addition to changes in lung ILC2s, the response of the lung tissue environment to HDM was also altered due to maternal asthma. We have included these findings in **Supplementary Fig. 1b** and added to the text (in lines 118-120). Upon re-analyzing the snRNA-seq data we had already performed, we found that the *Il33* gene was detected in alveolar type 2 cells (AT2), while the *Tslp* gene was found in pericytes and fibroblasts in the fetal lungs.

(A): Levels of IL-33, TSLP, and IL-25 in fetal lungs from asthmatic or control mothers. Fetal lungs were homogenized in 200 μ L of PBS, centrifuged, and the supernatants were analyzed by ELISA. (B): Dot plot showing the expression of *Il33*, *Tslp*, and *Il25* across different cell types in fetal lungs.

Fig 2e shows increased eosinophil recruitment of IL-33-treated mice given ILC2s from control or offspring of allergen-exposed moms. However, while the authors report that 5K of the ILC2s were given to recipients, it appears that >90% of the transferred cells are not making it to the lung (Supplementary Fig 1 (→Supplementary Fig 2c) shows only 200 – 400 ILCs in these lungs). Where do the majority of these cells go? Are the authors really comfortable arguing that there is no difference in cell numbers/recruitment given that they are only able to capture the recruitment of <10% of the input cells?

Thanks for the reviewer's thoughtful comment regarding the recruitment of ILC2s to the lung. We understand the reviewer's concern that not all transferred ILC2s are expected to establish residency in the lungs of recipient mice. However, the method of intravenously transferring sorted lung ILC2s has been validated in multiple previous studies, providing reproducibility in detecting ILC2s in the lungs of recipient mice (Halim TY, *et al.* Immunity. 2012, PMID: 22425247; Martinez-Gonzalez I, *et al.* Immunity. 2016, PMID: 27421705.; Xu H, *et al.* Nat Commun. 2023, PMID: 38097561). Additionally, a recent report by Xu H *et al.* demonstrated that the number of detected cells post-transfer was approximately one-tenth of the number transferred, which is consistent with our findings. It is important to note that the loss of cells may occur not only during the transfer process but also during the preparation of single-cell suspensions from the lung tissue of recipient mice.

I'm confused about the data shown in Supplementary Figure 2a and 2b (→Supplementary Figure 3a and 3b). The authors show that numbers of ILC2s in the lung do not differ in fetal mice given α IL-7R α , but that there are major differences in the numbers of ILC2s present in the lungs of mice 1 week after birth. If this is actually the case (i.e. I am not misunderstanding the data) that would strongly suggest that the typical expansion of ILC2 numbers after birth (or altered recruitment of a later "wave" of ILC2s to the lung) is really what is being impacted. How does this affect the overall interpretation of the authors' data?

Thanks for the reviewer's thoughtful comment regarding the data presented in Supplementary Figures 2a and 2b. We believe there may be a misunderstanding regarding the data. **Supplementary Figure 2a** (→Supplementary Figure 3a) does not represent data from the lungs of fetal mice given anti-IL-7R α . It shows results from another experiment using non-pregnant female mice in the OVA/Alum asthma model. The aim of this experiment was to confirm that the asthma in the OVA/Alum model mice was not alleviated by the anti-IL-7R α treatment. The data indicated that there were no significant differences in BAL eosinophil counts or lung ILC2 numbers between the anti-IL-7R α treated group and the isotype control

group in asthmatic mice. **Supplementary Figure 2b** (→Supplementary Figure 3b) demonstrates that the administration of anti-IL-7R α to asthmatic mothers leads to a significant decrease in ILC2 numbers in their offspring at the age of 1 week, as previously reported. We recognize that it is impossible to assess all effects on lung ILC2s during the period when they are absent. However, at least, the influence of changes in lung ILC2s during the fetal stage can be ruled out, considering that the administration of anti-IL-7R α during late pregnancy leads to the temporary deletion of ILC2s in the offspring. We acknowledge that the description of the experimental setup may have been unclear, and we have revised the manuscript text accordingly to improve clarity (in lines 175-180).

The authors demonstrate that the numbers of genes with altered expression in comparing populations of ILC2s from control and offspring of allergen-exposed moms is the largest, but there are significant differences in gene expression in other cell types (macrophages, epithelial cells). What are these differentially expressed genes? Is there any overlap in differentially expressed genes in ILC2s and these other populations?

Thanks for the reviewer's insightful question regarding the differentially expressed genes in macrophages and epithelial cells compared to ILC2s. We have conducted a more detailed reanalysis of the gene expression differences in macrophages and alveolar type 2 cells (AT2), including enrichment analysis. Our results indicate that in macrophages, there is an upregulation of genes associated with positive regulation of cell migration and leukocyte differentiation. In epithelial cells, particularly in AT2, we observed an increase in gene expression related to adherens junction and in utero embryonic development. Regarding overlap with ILC2s, we identified that leukocyte differentiation genes were shared with macrophages, while genes related to in utero embryonic development were common with AT2. These findings suggest that maternal asthma not only affects changes in ILC2s during the embryonic period but also impacts the differentiation and maturation of other immune cells, such as macrophages, as well as the development of epithelial cells, which are structural elements of the lung. We included the reanalysis results on the differentially expressed genes in macrophages and AT2 in **Supplementary Fig. 4h, i and Supplementary table 2**, and added to the text (in lines 225-234). Note that in **Supplementary Fig. 4a**, we selected the differential genes with large fold changes in the three cell types—ILC2, macrophages, and AT2—and compared their numbers. We also added this to the text (in lines 199-202).

The observation that corticosterone treatment of mothers can replicate the effect of allergen exposure is

interesting. Given that many asthmatic mothers are ALSO on steroids as a treatment for their allergic asthma, this may have profound influences on our understanding of transmission of allergic disease. Is application of both allergen AND steroids synergistic in terms of capacity to augment offspring asthma outcomes? The manuscript would be strengthened by performing such experiments. That the authors did not even mention this aspect of their study is surprising.

Thanks for the reviewer's insightful comment regarding the potential interactions between allergen exposure and corticosteroid treatment in asthmatic mothers. We induced chronic asthma in pregnant mice and concurrently administered oral dexamethasone (DEX) from E12 onwards. Our data indicate that there were no significant differences in lung eosinophil numbers between the adult offspring of mothers in the DEX combination group and those in the allergen-only group. While there were no major changes in cytokine-producing ILC2s, there was a tendency for an increased number of ILC2s in the adult offspring of mothers receiving DEX. Additionally, in another experiment using non-pregnant mice to assess the effects of DEX on our OVA/Alum asthma model, we found no significant differences in eosinophilic inflammation in the lungs between the two groups. These results suggest that, within the context of our experimental model, DEX treatment did not alleviate lung inflammation in the mothers, which likely reflects the lack of changes in eosinophil numbers in their adult offspring. However, we observed a trend towards an increase in lung ILC2 numbers in the offspring of mothers treated with DEX, indicating that systemic administration of steroids to asthmatic mothers may synergistically influence lung ILC2s in their adult offspring alongside the effects of asthma itself. These results have been added to **Supplementary Fig. 7c–e**, and corresponding text has been included in the manuscript (in lines 329-247).

Minor concerns:

The graphs in Fig 1b suggest that total CD45+ cells number $\sim 2 \times 10^6$, whereas B cells are close to 1×10^6 . This seems to be an awfully high number of B cells in the lung ($\sim 50\%$ of CD45+ cells). Is this a typo?

We rechecked the flow cytometry analysis with several colleagues and confirmed that there were no errors in the B cell counts. We based our protocol for preparing single-cell suspensions from lung tissue on previously published methods but made slight modifications to enhance the collection of ILC2s (e.g., changing the digestion solution every 20 minutes). This protocol may have influenced the higher number of B cells detected in the lung. To improve clarity, we made the labels for the numbers in the gating strategy

larger (**Supplementary Fig. 9**).

Please also show airway responses in the absolute (i.e. not normalized to baseline). It would be helpful to see the actual airway mechanic curves between the groups to allow the reader to better ascertain whether or not the effects of maternal allergen exposure are a result of altered baselines, or altered responsiveness to HDM.

In response to the reviewer's comment, we summarized the absolute values of the airway responses and included them in **Supplementary Table 1**. In addition, we generated airway mechanics curves based on the absolute values and included them in **Supplementary Fig. 1a** (in lines 116).

REVIEWER COMMENTS

Reviewer #2 (Remarks to the Author):

The Authors overall addressed my comments. It would be worth discussing one of the points in the revised data, however. The authors added a supplementary figure (Figure S7) illustrating no additional effects of DEX treatment in OVA-challenged mice. Since it is a clinically relevant question (maternal asthma treated by CS), I suggest discussing it further in the discussion section.

Thanks for the reviewer's beneficial comments. we have added further discussion on Figure S7 in the Discussion section (lines 449–460).